# Monocyte-driven atypical cytokine storm and aberrant neutrophil activation as key mediators of COVID-19 disease severity

L. Vanderbeke [1,26], P. Van Mol[2,26], Y. Van Herck [3,26], F. De Smet [4,26], S. Humblet-Baron[5,26], K. Martinod [6,26], A. Antoranz[4], I. Arijs [2], B. Boeckx[2], F. M. Bosisio[7], M. Casaer[8], D. Dauwe [8], W. De Wever[9], C. Dooms[10], E. Dreesen [11], A. Emmaneel [12], J. Filtjens[13], M. Gouwy[14], J. Gunst [8], G. Hermans[8], S. Jansen [15], K. Lagrou[1], A. Liston [16], N. Lorent[17], P. Meersseman[18], T. Mercier [1], J. Neyts [15], J. Odent[19], D. Panovska [4], P. A. Penttila [20], E. Pollet[19], P. Proost [14], J. Qian [2], K. Quintelier [12], J. Raes [21], S. Rex[22], Y. Saeys [12], J. Sprooten[23], S. Tejpar[24], D. Testelmans[10], K. Thevissen [25], T. Van Buyten[15], J. Vandenhaute[13], S. Van Gassen [12], L. C. Velásquez Pereira[6], R. Vos [10], B. Weynand[7], A. Wilmer[18], J. Yserbyt[10], A. D. Garg [23,27], P. Matthys [13,27], C. Wouters[5,13,27], D. Lambrechts [2,27], E. Wauters [10,27✉] & J. Wauters[18,27]

Epidemiological and clinical reports indicate that SARS-CoV-2 virulence hinges upon the triggering of an aberrant host immune response, more so than on direct virus-induced cellular damage. To elucidate the immunopathology underlying COVID-19 severity, we perform cytokine and multiplex immune profiling in COVID-19 patients. We show that hypercytokinemia in COVID-19 differs from the interferon-gamma-driven cytokine storm in macrophage activation syndrome, and is more pronounced in critical versus mild-moderate COVID-19. Systems modelling of cytokine levels paired with deep-immune profiling shows that classical monocytes drive this hyper-inflammatory phenotype and that a reduction in T-lymphocytes correlates with disease severity, with CD8+ cells being disproportionately affected. Antigen presenting machinery expression is also reduced in critical disease. Furthermore, we report that neutrophils contribute to disease severity and local tissue damage by amplification of hypercytokinemia and the formation of neutrophil extracellular traps. Together our findings suggest a myeloid-driven immunopathology, in which hyperactivated neutrophils and an ineffective adaptive immune system act as mediators of COVID-19 disease severity.

A full list of author affiliations appears at the end of the paper.

Since the first reports of a novel coronavirus outbreak in Hubei, China at the end of 2019, SARS-CoV-2 has rapidly spread across the globe, with more than 100 million people having been infected worldwide resulting in over 2 million fatal cases[1]. A broad spectrum of disease symptom manifestations and varied severity has complicated patient care and put enormous pressure on healthcare systems worldwide[2–7]. Our current understanding of the pathophysiology underlying distinct COVID-19 clinical phenotypes shows a clear link between the host immune response and disease severity. Early reports of fever, increased acute phase reactants and coagulopathy in severe COVID-19 cases combined with hypercytokinemia have pointed towards a so-called cytokine storm reminiscent of macrophage activation syndrome (MAS)[2,8,9]. Importantly, this paradigm has formed the basis for many interventional trials. However, it remains unclear how the dysregulated cytokine release observed in COVID-19 compares to other known cytokine storm syndromes[10,11].

It has become increasingly clear that the host response to COVID-19 differs substantially from the antiviral response to other respiratory viruses including influenza[12]. Decreased lymphocyte counts, reduced T-cell functionalities and increased neutrophil-to-lymphocyte ratio are now well-established hallmarks of COVID-19[13–21]. Despite a large-scale effort by the global research community over the past year, the exact mechanisms behind these innate and adaptive immune system alterations and their interplay remain to be fully elucidated. Based on results from plasma cytokine profiling, systems biology-driven predictive modelling and multiplexed immunophenotyping, here we show that (i) COVID-19 is characterized by an 'atypical' cytokine release with reduced type II interferon signalling, refuting the canonical cytokine storm paradigm, (ii) antigen presentation is impaired in critical disease and (iii) neutrophils are important effectors of the resulting damage both locally in the lung and systemically in circulation.

## Results

**Demographics and clinical data**. We prospectively recruited 61 hospitalized COVID-19 patients in our single-centre clinical study 'COvid-19 Advanced Genetic and Immunologic Sampling (COntAGIouS)', with demographics and clinical data summarized in Table 1. Routine clinical laboratory results at the time of study sampling are summarized in supplementary table 1. Of note, there was no significant difference in our patient population in baseline demographic characteristics related to disease severity, and all clinical findings are consistent with other published clinical cohorts of COVID-19 patients[2,22,23].

**Hypercytokinemia in COVID-19 as a distinct cytokine release syndrome**. A multitude of interventional trials are based on the paradigm of typical hypercytokinemia (such as in MAS) in critically ill COVID-19 patients, as a result thereby investigating immunomodulatory therapies such as anti-IL-6 or IFN-γ/IL-1 blockade. Surprisingly, a direct comparison between cytokine profiles in both disease entities has not been performed to date and scientific data on the contribution of these cytokines to COVID-19 is not unambiguous[10]. In an attempt to rationalise immunomodulatory treatment for critically ill COVID-19 patients, we have performed the first direct comparison between the plasma cytokine profiles in COVID-19 critical clinical condition and MAS (clinical characteristics of MAS cohort can be found in Suppl. Table 2).

Levels of IL-6, IL-10, IL-15, IFN-γ, TNF-α, CCL2, CCL3, CCL4, CCL26, CXCL9 and CXCL10 were significantly elevated both in COVID-19 critical clinical condition and in MAS patients as compared to healthy controls. For the majority of these cytokines and chemokines, including those that are typically associated with MAS (IL-6, IL-18, IFN-γ, TNF-α, CXCL9) the increase was less pronounced in COVID-19 critical condition than in MAS patients (Fig. 1a). On the other hand, some markers (i.e. IL-5, IL-7, IL-17A, CXCL8 and VEGF) were increased in critical COVID-19 patients only and not in MAS (Fig. 1a and Suppl. Fig. 1). Further comparison of the cytokine storm between critical COVID-19 and MAS revealed three key differences. A first feature distinct to COVID-19 is the relative absence of IFN-γ-associated cytokines and chemokines as compared to MAS patients. Indeed, levels of IL-18, a cytokine also known as IFN-γ-inducing factor, were significantly lower in critical COVID-19 than in MAS patients (Fig. 1a). Even more striking were the 50-fold lower plasma levels of IFN-γ in critical COVID-19 patients as compared to MAS. In line with reduced IFN-γ, we found markedly lower levels of CXCL9 (an IFN-γ-induced chemokine) in COVID-19 critical patients compared to MAS, indicative of decreased type II interferon signalling (Fig. 1a). IFN-γ is a central cytokine in cell-mediated immune responses that acts as a regulator of efficient antigen presentation and stimulator of cytotoxic T-lymphocytes (CTL), and its relatively low level is a key finding that differentiates critical COVID-19 hypercytokinemia from MAS cytokine storm. Secondly, levels of the main neutrophil chemoattractant CXCL8 were more than 3-fold higher in critical COVID-19 than in MAS, pointing toward a potential neutrophil signature (Fig. 1a). A third distinct feature of COVID-19 was the increased level of VEGF, indicating increased vascular modulation that is not evident in MAS[24]. In addition to these cytokine findings, we found acute phase reactants ferritin and D-dimers to be markedly lower in critical COVID-19 patients compared to MAS patients (Suppl. Table 1).

Next, we assessed whether this cytokine signature was restricted to critically ill COVID-19 patients, or present regardless of disease severity. A comparison between COVID-19 patients with critical versus mild-moderate clinical condition revealed significantly higher cytokine and chemoattractant levels (IL-5, IL-6, IL-7, IL-13, IL-15, IL-18, TNF-α, CCL2, CCL3, CCL4) in critical patients (Fig. 1b and Suppl. Fig. 2). Increased levels of IL-6 and TNF-α in critical patients could negatively affect the T-lymphocyte compartment, and indeed inversely correlated with clinical laboratory lymphocyte counts, with lymphocyte growth factors IL-7 and IL-15 suggesting a compensatory response. The chemokines CCL2 as well as CCL3 and CCL4 (also known as macrophage inflammatory protein 1-alpha and beta) are involved in the recruitment and activation of monocytes, macrophages and neutrophils. Plasma levels of other cytokines and chemokines, including IFN-γ, CXCL8 and CXCL9 were comparable between COVID-19 patients with critical and mild-moderate clinical condition (Fig. 1b and Suppl. Fig. 2).

Taken together, the plasma cytokine and chemokine profiles of COVID-19 described here are indicative of inflammatory myeloid cell and neutrophil involvement, which is most pronounced in critically ill patients. The compromised production of IFN-γ, a key cytokine in antigen presentation and development of adaptive immune responses, is striking in all COVID-19 patients regardless of disease severity.

**Classical monocytes orchestrate atypical COVID-19 cytokine release syndrome**. To identify the immune cells responsible for COVID-19 hypercytokinemia, we first performed unbiased immune prediction modelling using our experimental cytokine data. A (computational) predictive correlation network was constructed per immune cell type, based on expression profiles derived from 4639 human immune cell reference samples

**Table 1 Demographics and characteristics of patients infected with COVID-19.**

| | All patients (n = 61) | Mild-moderate clinical condition (n = 39) | Critical clinical condition (n = 22) | p-value |
|---|---|---|---|---|
| *Baseline characteristics* | | | | |
| Age, years | 62 [54–69] | 61 [56–69] | 63 [53–68] | 0.814 |
| Sex | | | | 0.173 |
| Men | 36 | 20 | 16 | |
| Women | 25 | 19 | 6 | |
| Comorbidity | | | | |
| Arterial hypertension | 32 (52) | 21 (54) | 11 (50) | 0.983 |
| Diabetes mellitus | 11 (18) | 7 (18) | 4 (18) | 1.000 |
| Chronic kidney failure | 8 (13) | 5 (13) | 3 (14) | 1.000 |
| Atrial fibrillation | 3 (5) | 3 (8) | 0 (0) | 0.547 |
| Obesity (BMI ≥ 30 kg/m²) | 20 (33) | 13 (33) | 7 (32) | 1.000 |
| Haematological malignancy | 1 (2) | 1 (3) | 0 (0) | 1.000 |
| Oncological malignancy | 6 (10) | 4 (10) | 2 (9) | 1.000 |
| *Clinical characteristics* | | | | |
| APACHE II | 16 [11–21] | NA | 16 [11–21] | - |
| Diagnosis of SARS-CoV-2 | | | | |
| CT compatible | 49 (80) | 29 (74) | 20 (91) | 0.182 |
| CT severity score | 9 (7–13) | 8 (5–10) | 13 (9–16) | **<0.0001** |
| qRT-PCR nasopharyngeal swab | 51 (84) | 33 (85) | 18 (82) | 1.000 |
| qRT-PCR BAL fluid | 6 (10) | 1 (3) | 5 (23) | **0.0198** |
| Respiratory support | 45 (74) | 23 (59) | 22 (100) | **0.0005[b]** **<0.0001[c]** |
| Oxygen via nasal cannula | 23 (38) | 23 (59) | 0 (0) | |
| High flow oxygen support | 14 (23) | 0 (0) | 14 (64) | |
| Invasive ventilation | 6 (10) | 0 (0) | 6 (27) | |
| Prone ventilation | 2 (3) | 0 (0) | 2 (9) | |
| Time from illness onset to sampling (days) | 9 [6–11] | 8 [5–11] | 10 [8–11] | 0.110 |
| Length of hospital stay (days) ((n)) | 11 [5–22] ((59)) | 6 [4–11] ((37)) | 24 [12–36] ((22)) | **<0.0001** |

Data are median [IQR], or n (%). The p-values comparing patients with mild-moderate and critical clinical condition are from Mann–Whitney U test[a] for continuous data, Cochran-Armitage test for trend[b] for ordered categorical data, and Pearson's $\chi^2$ or Fisher's exact test[c] for non-ordered categorical data, all based on a two-sided hypothesis. Chronic kidney failure was defined as eGFR <60 mL/min/1.73 m² during 3 months or structural renal disease under nephrology follow-up. Bold font is used to highlight statistically significant findings.
*BAL* bronchoalveolar lavage, *BMI* body mass index, *NA* not applicable.

assembled from 191 independently published studies[25]. This analysis creates a correlation network of input genes (in this case, genes coding for plasma cytokines detected in our COVID-19 study population), where only those genes that have a high probability of co-expression (based on extensive immune cells' reference-gene profiles) are connected. This reveals a high probability for these genes to associate with overlapping immuno-regulatory modules specific for each immune cell type. As expected, our predictions pointed towards dominant myeloid-driven inflammation in COVID-19, with most extensive correlation networks of cytokine and chemokine-coding gene profiles formed within macrophages and neutrophils, and limited predictions within various lymphocytes. Interestingly, mild-moderate COVID-19 patients could be distinguished from critical COVID-19 patients by a higher cytokine/chemokine-coding gene connectivity in plasmacytoid dendritic cells (DCs) and particularly reduced connectivity in neutrophils (Suppl. Fig. 3a).

To validate the hypothesis of myeloid-driven immunopathology generated by these quantitative cytokine and qualitative computational immunology analyses, we subsequently performed immunophenotyping experiments on circulating immune cells using three complementary techniques. Mass cytometry of whole blood was used to quantify and profile general leucocyte cell types including peripheral blood mononuclear cells (PBMCs) and granulocytes, while scRNA-seq of PBMCs allowed for their characterization at the RNA level and identified cellular sources of cytokine expression. In addition, classical flow cytometry after PMA/ionomycin stimulation of lymphocytes isolated from PBMCs allowed functional characterization of the adaptive immune system. Overall, mass cytometry showed a decrease of most immune populations in COVID-19 patients compared to healthy controls, with more pronounced reductions in the critical than in the mild-moderate group; a finding in line with clinical reports[2,22,23]. However, contradicting the paradigm of overall immune cell decrease in COVID-19, plasmablast and neutrophil counts increased compared to healthy controls, more so in patients in critical as compared to in mild-moderate condition (Fig. 2a).

Substantiating our hypotheses on the cellular origin of cytokine production, scRNA-seq of PBMCs showed that classical mono-cytes were the main source of major COVID-19 mediating cytokines, including the monocyte chemoattractant CCL2 and its receptor CCR2, the neutrophil chemoattractant CXCL8, and TNF-α. The pro-inflammatory nature of this cell subset is further highlighted by its high expression of inflammasome-associated cytokine coding genes *IL1B* and *IL18* (Fig. 2b, c). Furthermore, although monocytes showed the most notable drop in cell count when comparing critical to mild-moderate condition evaluated by CyTOF and scRNA-seq, the latter revealed a significant decrease in non-classical monocytes (based on *C1AQ*, *C1BQ* and *LSTB1* marker expression) and a corresponding relative increase of classical monocytes (based on *S100A8*, *S100A9* and *S100A12* marker expression) in critical COVID-19 (Fig. 2a and Suppl. Fig. 4d). Classical monocytes have previously been shown to have a pro-inflammatory phenotype, with non-classical monocytes being recognized as patrolling phagocytosing and anti-inflammatory cells that play an important role in antiviral defence[26,27]. Moreover, classical monocytes show a higher

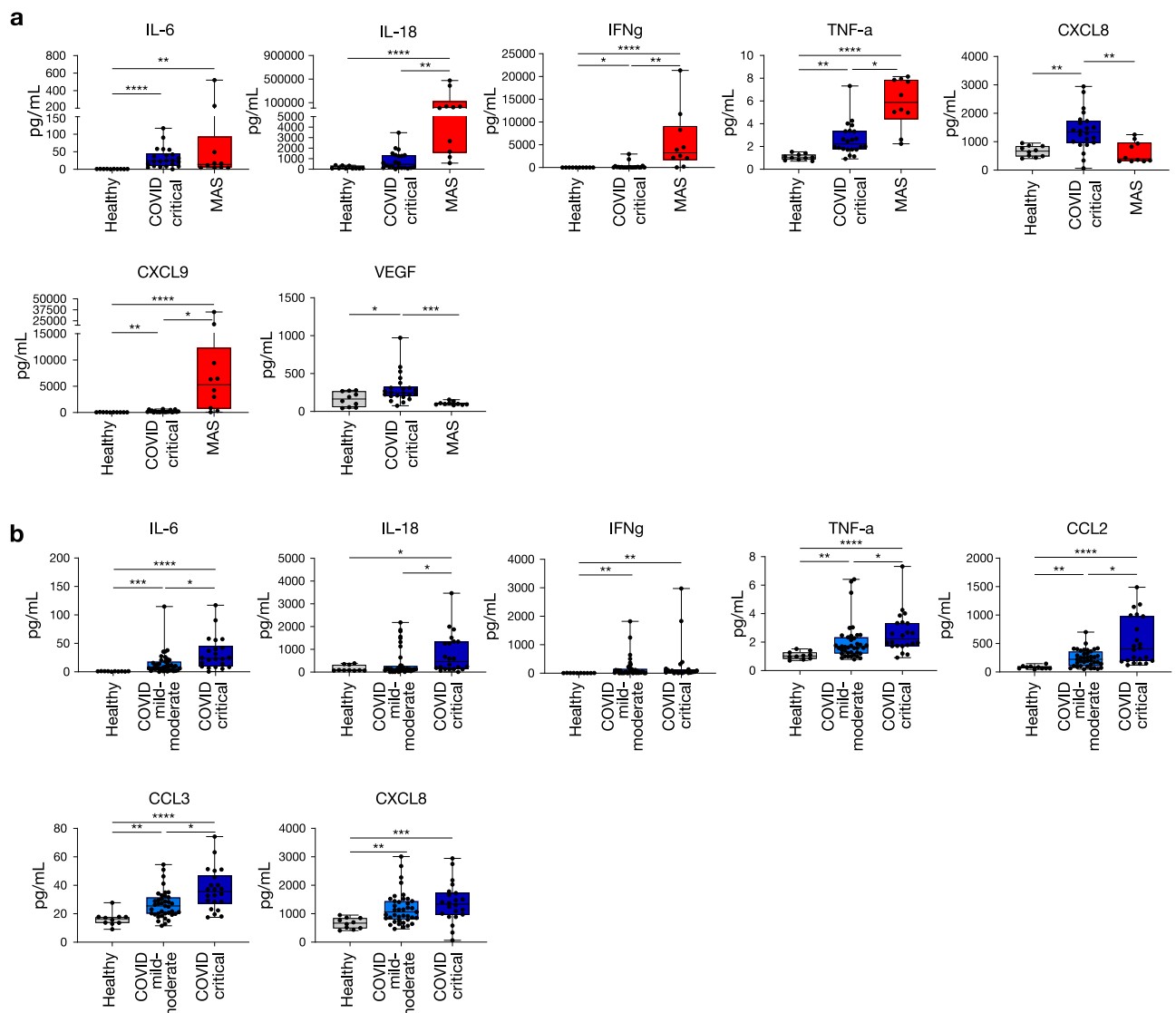

**Fig. 1 Hypercytokinemia in COVID-19 as a distinct cytokine release syndrome.** Comparison of plasma levels of selected cytokines and chemokines from healthy controls (HC, n = 10), COVID-19 critical condition (CCC, n = 22) and MAS patients (n = 10) (**a**) and COVID-19 subgroups (for mild-moderate (CMM), n = 39; for critical (CCC), n = 22) versus healthy controls (n = 10) (**b**). Plasma concentrations were measured by MSD (Meso Scale Discovery). Boxplot representation (centre line, mean; box limits, upper and lower quartiles; whiskers, range; points, data points per patient). A two-sided Kruskal–Wallis test with Dunn's correction for multiple comparisons was used; IL-6: p < 0.0001 HC vs CCC, p = 0.003 HC vs MAS; IL-18: p < 0.0001 HC vs MAS, p = 0.005 CCC vs MAS; IFNg: p = 0.014 HC vs CCC, p < 0.0001 HC vs MAS, p = 0.008 CCC vs MAS; TNF-a: p = 0.004 HC vs CCC, p < 0.0001 HC vs MAS, p = 0.015 CCC vs MAS; CXCL8: p = 0.008 HC vs CCC, p = 0.002 CCC vs MAS; CXCL9: p = 0.007 HC vs CCC, p < 0.0001 HC vs MAS, p = 0.014 CCC vs MAS; VEGF: p = 0.049 HC vs CCC, p = 0.0004 CCC vs MAS (**a**) and IL-6: p = 0.0009 HC vs CMM, p < 0.0001 HC vs CCC, p = 0.014 CMM vs CCC; IL-18: p = 0.019 HC vs CCC, p = 0.024 CMM vs CCC; IFNg: p = 0.004 HC vs CMM, p = 0.003 HC vs CCC; TNF-a: p = 0.009 HC vs CMM, p < 0.0001 HC vs CCC, p = 0.038 CMM vs CCC; CCL2: p = 0.003 HC vs CMM, p < 0.0001 HC vs CCC, p = 0.026 CMM vs CCC; CCL3: p = 0.006 HC vs CMM, p < 0.0001 HC vs CCC, p = 0.016 CMM vs CCC; CXCL8: p = 0.007 HC vs CMM, p = 0.0005 HC vs CCC (**b**). Significance is shown as *p < 0.05; **p < 0.01; ***p < 0.001 and ****p < 0.0001. MAS = macrophage activation syndrome. See Figs. S1 and S2 for additional cytokine results. Source data are provided as a Source data file.

expression of the monocyte chemoattractant *CCR2* compared to non-classical monocytes, with anti-CCR2 treatment ameliorating SARS-CoV-1 disease course in preclinical models[28]. The importance of non-classical monocyte depletion in COVID-19 immunopathology has been demonstrated in an independent cohort of patients with COVID-19, as evidenced by recovery of this cell population during later stages of disease[19].

Interferon-γ was, expectedly, predominantly found in cytotoxic CD8+ T and NK cells, and to a lesser extent in CD4+ T cells (Fig. 2c). These immune cell populations were shown to be globally decreased in COVID-19 compared to healthy controls

(Fig. 2a). However, an increased CD4+/CD8+ T-cell ratio was seen, which was most pronounced in the critical disease group. This contrasts with typical findings during viral respiratory infection, such as influenza or SARS, or sepsis where lymphocytopenia is evident but CD4+/CD8+ ratios are decreased[29–31]. At the single-cell RNA-seq level, the increased CD4+/CD8+ T-cell ratio was confirmed when comparing COVID-19 patients to healthy controls (Fig. 2d).

We further explored specific CD4+ and CD8+ T-cell subsets using our flow cytometry data (publicly available at https://flowrepository.org/experiments/2713). In line with Neumann

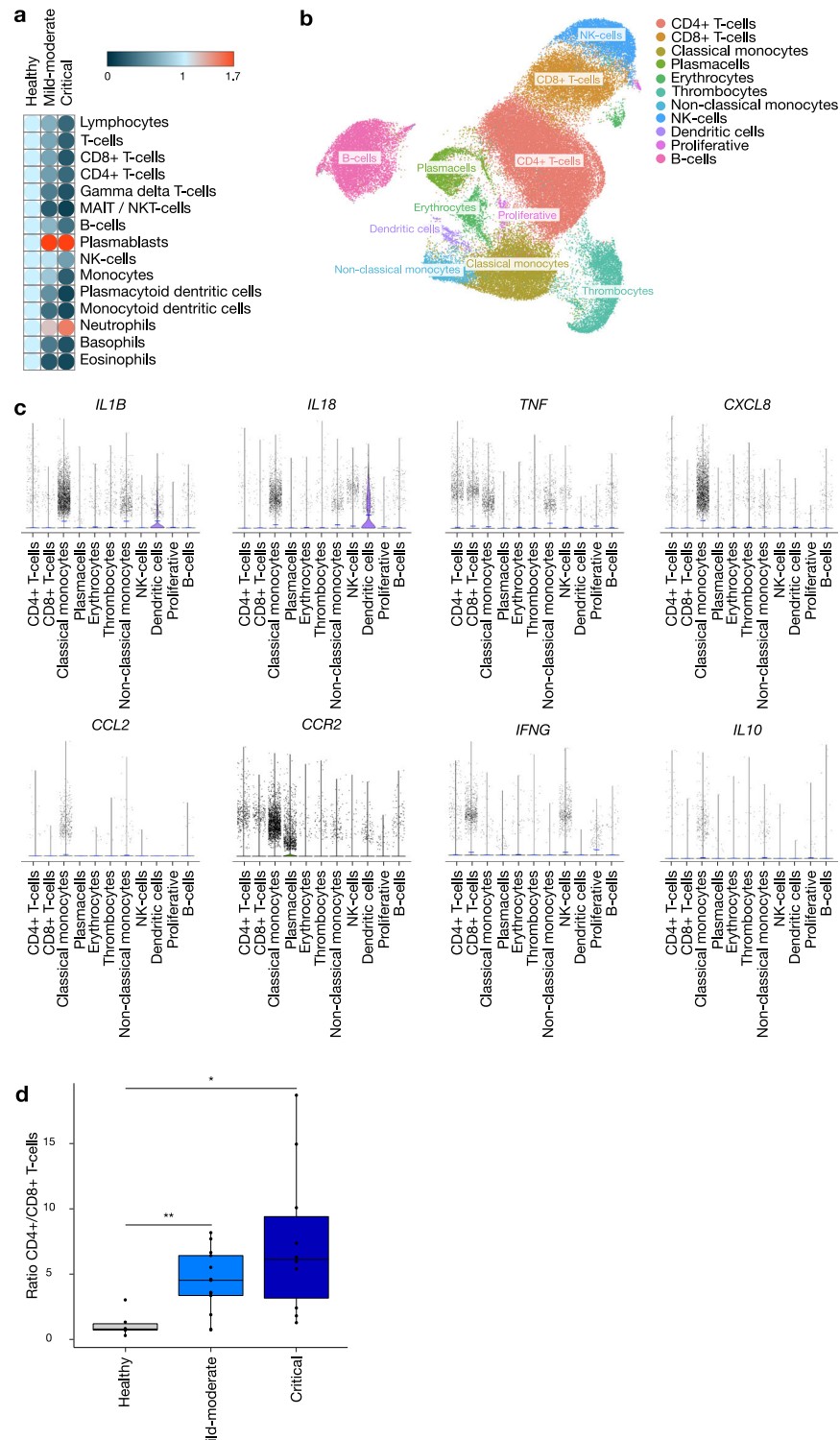

**Fig. 2 Peripheral blood immunophenotyping. a** Heatmap representation of immune cell subset changes between healthy controls ($n = 8$), COVID-19 mild-moderate ($n = 32$) and critical ($n = 14$) condition based on mass cytometry measurements on whole blood; representation shows relative fold change compared to healthy per cell subset. A two-sided Wilcoxon rank-sum test with Benjamini–Hochberg correction for multiple group comparisons was used. **b** scRNA-seq data of COVID-19 PBMCs: UMAP plot of 83,524 single cells, colour-coded per cell type. **c** Violin plots of expression level of key cytokine/chemokine/chemokine receptor coding genes in the cell types identified in PBMCs from COVID-19 patients, as shown in (**b**). **d** CD4+/CD8+ ratios in healthy controls (HC, $n = 6$), mild-moderate (CMM, $n = 13$) and critical (CCC, $n = 10$) COVID-19 cases, based on scRNA-seq. Healthy control data were derived from a publicly available dataset (GSE150728). Boxplot representation (centre line, mean; box limits, upper and lower quartiles; whiskers, range; points, data points per patient). A two-sided Wilcoxon rank-sum test with Benjamini–Hochberg correction for multiple group comparisons was used. $p = 0.002$ HC vs CMM, $p = 0.011$ HC vs CCC. Significance is shown as *$p < 0.05$; **$p < 0.01$. See Figs. S4 and S5 for further supporting data. Source data are provided as a Source data file.

et al.[32], there was no significant difference in T-helper and cytotoxic T-cell counts between the COVID-19 disease severity groups. However, we found the T-helper 1 subset (defined as CD4+ T cells secreting IFN-γ) significantly decreased when comparing all COVID-19 patients to healthy controls. Interestingly, PD-1 was highly expressed in most of the effector T cells with a predominance in CD8+ TEMRA cells, attesting for the higher activation of these cells without a strong Th1/Tc1 polarisation (Suppl. Fig. 6a, b). Taken together, the relatively low IFN-γ signalling identified in COVID-19 could be explained by reduced lymphocyte counts, with CD8+ T cells most affected in critical condition.

**Reduced MHC class II on antigen-presenting cells marks critical COVID-19.** Having identified (quantitative) shifts in immune cell populations underlying cytokine changes seen in COVID-19, we looked at putative downstream effects of these cytokine alterations. Specifically, given the relatively reduced IFN-γ signalling and the importance of this cytokine in antigen presentation, we analysed presence of relevant antigen presentation molecules in COVID-19 by mass and flow cytometry. When comparing monocyte populations between critical and mild-moderate COVID-19, mass cytometry results indeed revealed an overall shift towards reduced expression of molecules typically involved in antigen presentation (Fig. 3a). Further evidence was obtained by flow cytometric analysis that showed significantly decreased intensity of HLA-DR staining on CD14$^{hi}$ monocytes in COVID-19 patients compared to healthy controls (Fig. 3b). A similar trend was seen in classical monocytes using scRNA-seq, though not reaching statistical significance ($p = 0.06$; Fig. 3c).

Next, we investigated whether antigen-presenting capacity was also reduced in 'true' professional antigen-presenting cells (APCs). Mass cytometric analysis showed a significant decrease in HLA-DR on myeloid dendritic cells from critical compared to mild-moderate COVID-19 patients (Fig. 3d). Single-cell RNA-seq confirmed significant downregulation of genes encoding HLA-DR on dendritic cells in critically ill patients (Fig. 3c). Moreover, expression of co-stimulatory factors *CD83*, *CD86*, *ICOSLG* and *ICAM1* on dendritic cells was also decreased in critical patients (Fig. 3e). Given the inflammatory context, these findings strongly suggest that antigen-presenting capacity of dendritic cells is differentially affected in critical COVID-19 cases.

**Disturbed immuno-regulation in severe COVID-19.** In order to evaluate relationships between cytokine and immune cell shifts and assess qualitative differences between mild-moderate and critical COVID-19 patients in an unbiased fashion, we performed similarity matrix-based statistical correlation modelling analyses (Pearson correlation-driven) to decipher statistically-stable clustering patterns between plasma-screening-derived cytokines/chemokines and mass cytometry-derived whole-blood peripheral immune cell enrichments (Fig. 4a, b).

At the level of cytokines/chemokines, patients with critical COVID-19, compared to mild-moderate COVID-19 patients, showed a tendency to gain a highly correlated (and expanded) cluster of IL-1 cytokines as well as a tendency of an IFN-γ/IL-6 cluster to gain correlation with TNF-α (Fig. 4a, b). Overall, this points to a highly pro-inflammatory cytokine co-association profile as a distinguishing characteristic of COVID-19 disease severity.

Within the lymphoid compartment, mild-moderate COVID-19 patients had better correlation between CD4+/CD8+ T cells and some B-cell subsets, whereas in critical COVID-19 patients such correlations between these lymphocyte subpopulations became more fragmented. Specifically, CD4+ T cells and CD8+ T cells

did not exhibit a tendency to correlate with each other, which can cause major dysregulation of lymphocyte functioning and communication. These trends highlight the disparities in qualitative functional cross-talk within the lymphoid compartment of critical COVID-19 patients (Fig. 4a, b). In line, network analyses of cytokines correlating with these lymphocytes (while integrating Gene Ontology/GO-based immunological biological processes) showed that, cytokines correlating with CD4+/CD8+/B-lymphocytes in mild-moderate COVID-19 patients (e.g. IL-16) had a clear pro-effector orientation (enriching for GO terms for effector, activation or defense response functions), whereas cytokines correlating with these lymphocytes in critical COVID-19 patients (e.g. IL-1α, CCL22, CCL17) had characteristics of activation-associated stress/cell death (enriching for GO terms for defense response and also cell death or cell stress) (Suppl. Fig. 3b).

Unlike most lymphocytes, various myeloid cells showed differential tendencies to positively correlate with different cytokines; especially plasmacytoid DCs and neutrophils. However, mild-moderate COVID-19 patients had better correlation between neutrophils and NK-/Th17 cells, whereas in critical COVID-19 patients, such correlations became more fragmented. As such, neutrophils gained considerable correlation with specific cytokines in critical patients and did not correlate sufficiently with NK-/Th17 cells—this possibly suggests disruption of an immune-regulatory loop (where Th17-neutrophil cross-talk can enable a more controlled or immune-regulated inflammatory reaction) and may indicate unleashing of detrimental neutrophil-based inflammation[33]. Network analyses of cytokines correlating with these neutrophil-based clusters reinforced this hypothesis (Suppl. Fig. 3b). Here, we noted that the GO biological process terms most often enriched by cytokines that correlate with neutrophil clusters in mild-moderate COVID-19 patients, mainly included "defense response", "positive regulation of immune system", "cell motility" and "leucocyte migration", a sign of a normal innate immune response. A similar analysis in critical COVID-19 patients showed enrichment of GO biological process terms similar to the ones stated above, but further combined with "regulation of programmed cell death". This suggested the possibility of increased cellular stress or cell death in neutrophils brought about by increased inflammation.

Overall, these computational systems and network biology assessments predicted that mild-moderate COVID-19 patients could be characterized by more functional/regulated immunopathology (i.e. effector lymphocyte cross-talk and immune-regulated inflammatory reactions), whereas critical COVID-19 patients might be marked by dysfunctional/dysregulated immunopathology. The latter is immunologically distinguished by dysregulation of the lymphoid compartment (e.g. fragmented cross-talk and signs of inflammation-induced stress) and extremely high (unregulated) neutrophil-driven inflammation.

**Neutrophil extracellular trap formation in severe COVID-19.** The above predictions clearly pointed to a qualitatively differential immunological activity of neutrophils as one of the most important determinants of critical COVID-19 immunopathology. This urged us to perform more in-depth (quantitative and functional) evaluation of this immune subset.

First, we examined neutrophil activation status and neutrophil extracellular trap (NET) formation in COVID-19 as compared to healthy subjects, by assessing myeloperoxidase (MPO), MPO-DNA and citrullinated histone H3 levels, respectively[34] (Suppl. Fig. 7a–c). We additionally compared these levels between COVID-19 patients and patients with non-COVID pneumonia (all hospitalised, non-ventilated patients; for clinical characteristics see Suppl. Table 2), showing significantly higher neutrophil

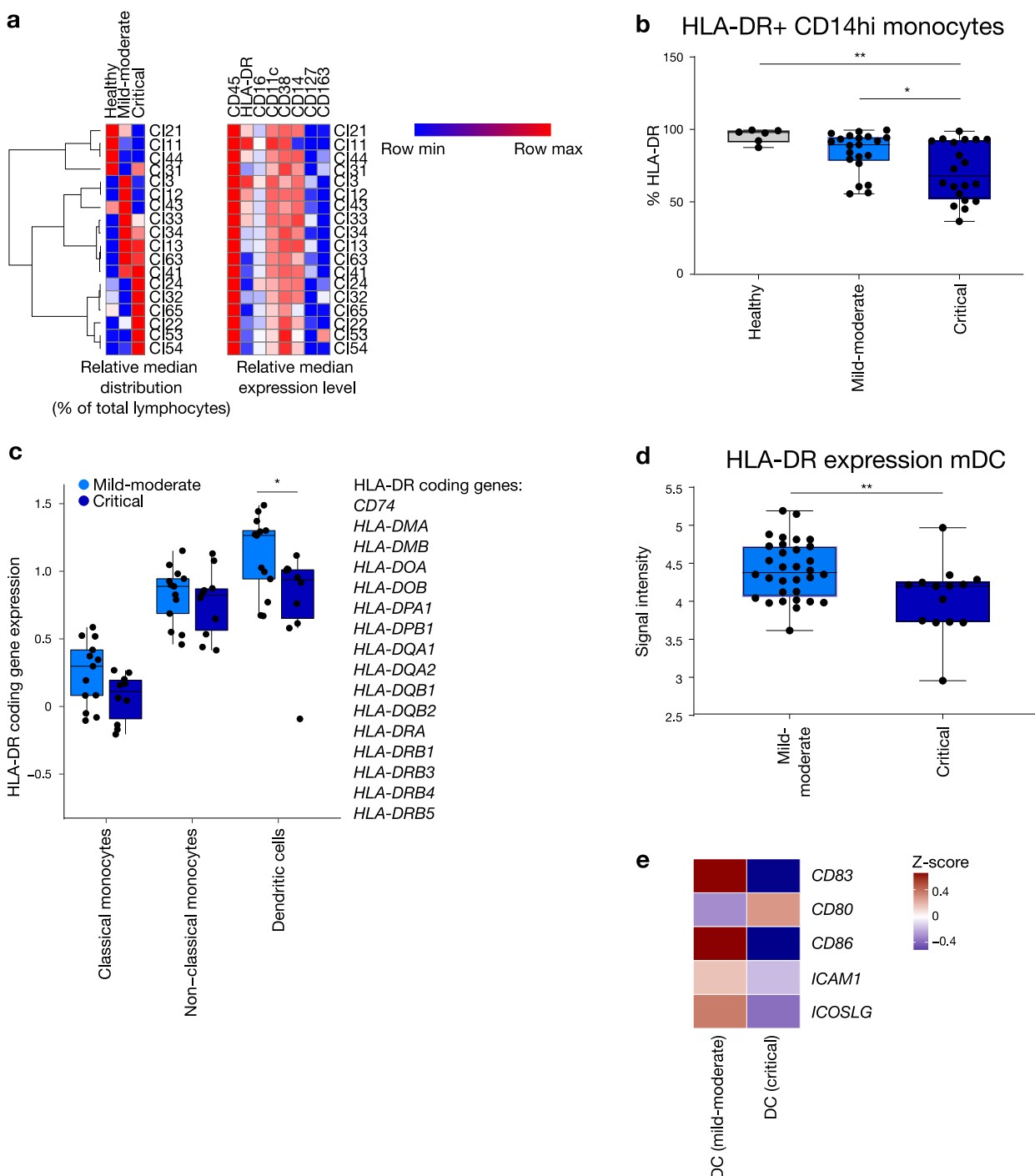

**Fig. 3 Reduced MHC class II on antigen-presenting cells marks critical COVID-19. a** Heatmap of differential monocyte clusters between healthy ($n = 8$), mild-moderate ($n = 32$) and critical ($n = 14$) COVID-19 groups based on mass cytometry measurements on whole blood. Rows indicate monocyte subclusters. Columns indicate patient groups (left) and cell surface markers (right). A two-sided Wilcoxon rank-sum test with Benjamini–Hochberg correction for multiple group comparisons was used. **b** HLA-DR expression on CD14$^{hi}$ monocytes in healthy (HC, $n = 6$; mean 95,72%), mild-moderate (CMM, $n = 21$; mean 84,14%) and critical (CCC, $n = 20$; mean 70,97%) groups based on flow cytometric analyses of PBMCs. Boxplot representation (centre line, mean; box limits, upper and lower quartiles; whiskers, range; points, data points per patient). A two-sided Wilcoxon rank-sum test with Benjamini–Hochberg correction for multiple group comparisons was used; $p = 0.0014$ HC vs CCC, $p = 0.046$ CMM vs CCC. **c** Gene set enrichment analysis of HLA-DR complex coding genes in professional antigen-presenting cells, comparing scRNA-seq data from mild-moderate ($n = 13$) and critical ($n = 10$) COVID-19 cases. Boxplot representation (centre line, mean; box limits, upper and lower quartiles; whiskers, range; points, data points per patient). A two-sided Wilcoxon rank-sum test was used, $p = 0.0303$ for dendritic cells (DC). **d** Expression of HLA-DR on myeloid DC based on mass cytometry measurements (mild-moderate $n = 32$, critical $n = 14$). Boxplot representation (centre line, mean; box limits, upper and lower quartiles; whiskers, range; points, data points per patient). A two-sided Wilcoxon rank-sum test was used, $p = 0.0098$. **e** Heatmap of genes coding for co-stimulatory molecules involved in MHC class II-restricted antigen presentation by dendritic cells, comparing mild-moderate versus critical COVID-19. Individual dots in boxplots represent data points per patient. Source data are provided as a Source data file. Statistical significance is shown as *$p < 0.05$; **$p < 0.01$.

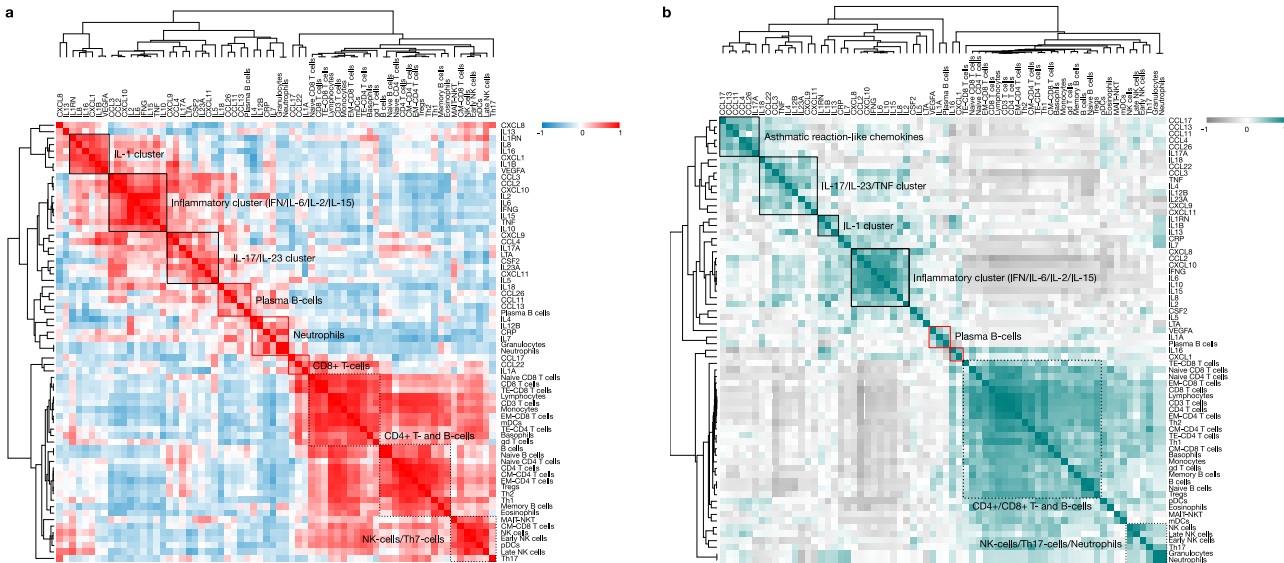

**Fig. 4 Disturbed immune regulation in severe COVID-19.** Pearson correlation-driven similarity/correlation matrix analysis of cytokines/chemokines and mass cytometry data in critical ($n = 14$) (**a**) and mild-moderate ($n = 31$) (**b**) COVID-19 patient subgroups. This correlation matrix analysis is a form of statistical modelling by which statistically stable relationships between the different variables (i.e. cytokines/chemokines and immune cell subpopulations) allows their categorization into different clusters indicating high levels of correlation (indicated by the clustering dendrograms). Of note, the diagonal correlation value is 1, which denotes the highest possible statistically significant correlation value between the given variables and represents the highest comparative threshold that "centers" the correlation network. See also Fig. S3. Source data are provided as a Source data file.

activation and circulating NETs in the COVID-19 cohort (Suppl. Fig. 7d–f). Within the COVID-19 cohort, the highest levels were noted in the critical disease group (Suppl. Fig. 7g–i). Overall, these data show that, besides an increase in neutrophil counts, there is increased neutrophil activation and NET release in COVID-19 linked to disease severity. Importantly, these evaluations were performed in plasma, clearly indicating that circulating NETs contribute to systemic inflammatory responses and this is likely not resulting from mechanical ventilation-induced pulmonary stress.

Based on these results and previously published autopsy reports, we aimed to confirm that hyperactivated neutrophils not only play an important role in systemic COVID-19 disease manifestations, but also contribute to severe COVID-19 pneumonia[35,36]. We therefore performed scRNA-seq on fresh BAL fluid from 6 COVID-19 and 5 non-COVID pneumonia cases, sequencing the transcriptomes of 26,605 cells in total (Fig. 5a and Suppl. Fig. 8a–c). We observed a striking enrichment of neutrophils in the lungs of COVID-19 patients compared to non-COVID pneumonia cases (Fig. 5e). Moreover, a much larger proportion of these neutrophils was in a hyperactivated state in COVID-19 patients, marked by upregulated *IL1B*, *CXCL8* and *S100A12* expression (Fig. 5b–d). In-depth differential gene-expression analysis revealed upregulation of other activation markers (e.g. *S100A8, S100A9, FPR1, SOD2*) as well as inflammasome-stimulating genes (*NEAT1*)[37]. Intriguingly, so-called resting neutrophils showed relative upregulation of genes coding for HLA-DR receptors (Suppl. Fig. 8f). Although neutrophils are typically considered poor antigen-presenters, our findings suggest that, similar to what we observed in circulating myeloid cells, 'active' neutrophils in the lungs of critical COVID-19 patients have further lost antigen-presenting capacity and have adopted a deleterious phenotype that contributes to inflammatory cytokine signalling (*IL1B*, *CXCL8*, *NEAT1*) and local tissue damage (*FPR1*).

To investigate NET formation as a component of neutrophil-induced lung tissue damage in severe COVID-19, we selected 18 genes with an established role in NET formation, adapted from

Gardinassi et al.[38], and then calculated a NET score based on the average up- or downregulation of their expression. This score was significantly higher in the active neutrophil population (Suppl. Fig. 8g). Based on this, we suggest that hyperactivated neutrophils and resulting NET formation not only contribute to systemic illness, but also to lung damage in severe COVID-19.

## Discussion

In this prospective, case-control study of 61 patients with varying degrees of COVID-19 disease severity and 31 control patients, we used a quantitative and integrative qualitative immunopheno-typing approach to characterize the cytokine responses in COVID-19, the upstream mechanisms and downstream effects with emphasis on their impact on disease severity. Our results identified a myeloid-driven atypical cytokine storm that is distinctly different from MAS, with specific contributions of classical pro-inflammatory monocytes and neutrophils dominating COVID-19 immunopathology in the most critical cases.

The first line of an effective antiviral defense consists of immune sensing of viral RNA by pattern recognition receptors (PRR) on innate immune and epithelial cells. Downstream signalling results in early IFN type I–III secretion, which raises an adaptive immune response and recruits specific leucocyte subsets to the site of inflammation. These leucocytes create a pro-inflammatory milieu (through secretion of IL-6, IL-8 and TNF-α amongst others) later in the disease course[3,39,40]. Current state of the art shows that the host innate immune response in COVID-19 patients is both ineffective at timely stimulation of the adaptive immune system, attributed to a delayed type I and type III IFN signature, and excessive causing local (lung) tissue damage and a systemic cytokine storm with fever, increased acute phase reactants and multiple organ involvement in severely affected cases[2,14,40–42]. Importantly, Galani et al. have shown that the pathogenicity of influenza virus, a major cause of severe viral pneumonia, does not depend on disturbing this antiviral cascade[40]. How exactly pathogenic coronaviruses trigger this imbalanced myeloid activation, which is very pronounced in

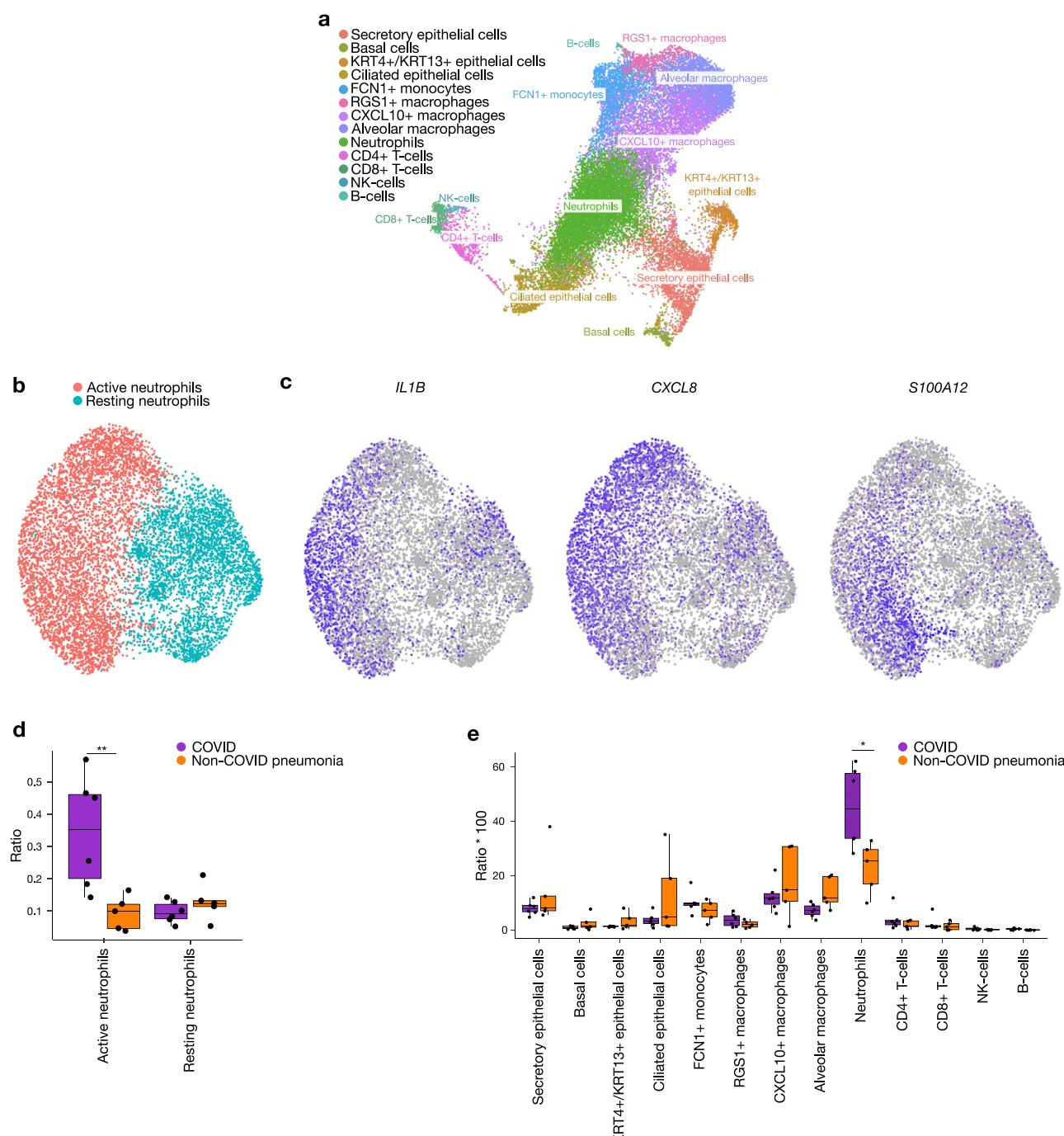

**Fig. 5 Contribution of neutrophils to COVID-19 immunopathology at a local level: scRNA-seq data of COVID-19 BAL fluid.** UMAP plot of 26,605 single cells (from 11 patients, $n = 6$ for COVID-19, $n = 5$ for non-COVID pneumonia), colour-coded per cell type (**a**) and UMAP showing active and resting neutrophil subclusters (**b**) present in the bronchoalveolar lavage fluid of COVID-19 and non-COVID pneumonia patients. **c** Feature plots of key differentially expressed genes, with *IL1B, CXCL8* and *S100A12* being upregulated in the active neutrophil population. **d** Boxplots showing a significant abundance of active neutrophils in COVID-19 pneumonitis, as compared to non-COVID pneumonia. Boxplot representation (centre line, mean; box limits, upper and lower quartiles; whiskers, range; points, data points per patient). A two-sided Wilcoxon rank-sum test was used, $p = 0.009$ active neutrophils COVID vs Non-COVID pneumonia. **e** Relative immune cell type abundance in bronchoalveolar lavage fluid of COVID-19, compared to non-COVID pneumonia. Boxplot representation (centre line, mean; box limits, upper and lower quartiles; whiskers, range; points, data points per patient). A two-sided Wilcoxon rank-sum test was used, $p = 0.017$ neutrophils COVID vs non-Covid pneumonia. Source data are provided as a Source data file. Significance is shown as *$p < 0.05$; **$p < 0.01$. See also Fig. S8.

COVID-19, remains a matter of debate. Mechanisms of immune evasion by SARS-CoV-2 have extensively been documented by Lei et al.[43]. The resulting delayed IFN signalling and sustained viral replication might promote cytolysis. This in turn might recruit immune cells and induce their cytokine production in a feedforward-loop manner[40,44]. A direct effect of SARS-CoV-2 proteins on cytokine production by epithelial and myeloid cells was put forward in an elegant proteomic study, suggesting that SARS-CoV-2 nsp9 and nsp10 can interfere with NKRF (an NF-kB repressor), thereby stimulating IL-6 and IL-8 production[39,45].

SARS-CoV and SARS-CoV-2, but not MERS-CoV, binding to and subsequent downregulation of ACE2 contributes to this pro-inflammatory signalling[46,47].

Excessive cytokine release is seen as a key driver of immunopathology in critically ill COVID-19 patients, and many interventional immunomodulatory trials are underway targeting MAS-like cytokine signalling pathways, despite the fact that their similarity has never been formally investigated[10,11]. We performed, to our knowledge, the first direct comparison between COVID-19 critical illness and the typical cytokine release syndrome MAS. Although we identified some parallels, the dysregulated cytokine release in COVID-19 was shown here to be distinctly different from that in MAS[2,48,49]. Most strikingly, we found markedly reduced type II IFN signalling, which was also evident in a comparison of SARS-CoV-2 and influenza as published by Mudd et al.[50]. This highlights the significance of this cytokine in COVID-19 immunopathology. In addition, despite defective interferon production after SARS-CoV-2 infection, the presence of highly elevated monocyte and neutrophil chemoattractants advocates a strong case for myeloid-driven innate immune hyperactivity in COVID-19. Lastly, vascular remodelling, as suggested by elevated VEGF levels in the systemic circulation, further distinguishes COVID-19 from MAS hypercytokinemia, findings consistent with autopsy reports on immunothrombosis and microcoagulopathy[35,51]. Our results support the concept that "cytokine storm" is too broad a term to be used for the many types of hypercytokinemia, which was recently corroborated by a comparison of IL-1β, IL-6, IL-8 and TNF-α levels in COVID-19 and CAR T-cell therapy induced cytokine release[52].

We had hypothesized that a major myeloid cell-driven footprint would be underlying these cytokine changes, and we systematically confirmed this by multiplexed immunophenotyping analyses. Here, we identified a pro-inflammatory monocyte subset, the classical monocytes, as the most important source of inflammatory cytokines within PBMCs collected from COVID-19 patients. Their relative increase and cytokine production paired to a decrease of non-classical monocytes, an anti-inflammatory myeloid cell type, is a key determinant of severe COVID-19; now a well-substantiated finding in the COVID-19 literature[53,54] (although not confirmed in another recently published small transcriptomic cohort[55]).

At the other end of this hyperinflammatory innate immune response, lies a defective virus-directed response of the adaptive immune system. Because IFN-γ is a key factor enabling efficient antigen presentation via MHC class II molecules, we speculated adaptive immune priming would be impaired in COVID-19. Impaired antigen presentation by CD14+ monocytes in COVID-19 has previously been suggested, mediated by IL-6[5]. Importantly, we showed the antigen presentation pathway, including co-stimulatory receptors, to be even more affected in 'true' professional antigen-presenting cells (i.e. dendritic cells). As such, the host likely fails to mount an adequate adaptive antiviral immune response, in line with published flow cytometry data[56]. In addition to this putative reduced T-cell function, there is a clear global quantitative reduction of T-lymphocytes in COVID-19, related to disease severity and inversely related to levels of IL-6 and TNF-α (as previously reported[14]). Importantly, CD8+ T cells were more affected, leading to an increased CD4+/CD8+ T-cell ratio especially in critical disease. This distinguishes COVID-19 from other viral respiratory infections and bacterial sepsis, where lymphocytopenia is evident but CD4+/CD8+ ratios are decreased[29–31]. This could explain the relatively reduced IFN-γ levels we observed; a hypothesis that is reinforced by our single-cell RNA-seq data.

Finally, our computational prediction analyses suggested that neutrophils could be important effector cells for distinguishing mild-moderate from critical disease. We confirmed this experimentally by determining surrogate neutrophil activation and NET forming activity in the circulation, and showed these were elevated in COVID-19, particularly with increased disease severity. Our findings are in line with recent publications[34,57,58], with the addition of a non-COVID pneumonia control group indicating COVID-specificity for these high NET levels in our hospitalized cohort. Mechanical ventilation has been shown to induce NET formation in the alveolar space[59], yet our analyses were performed in plasma, evidencing the importance of this process in systemic COVID-19 immunopathology. Moreover, our scRNA-seq data showed activated neutrophils to be key effectors, not only of systemic inflammation, but also of lung damage in severe COVID-19. Indeed, lung neutrophils have a highly activated phenotype and show upregulation of NET formation related genes. Formation of NETs in the lungs and bloodstream will have clear pathological consequences, inducing epithelial/endothelial cell death, contributing to thrombotic complications, and potentially having future implications for fibrotic remodelling in the lung[60].

This study has some limitations. First, our study cohort only includes COVID-19 cases requiring hospitalisation in our tertiary care centre. Thus, it does not cover the entire disease severity spectrum. Second, all samples were intentionally collected at the same timepoint, as the immune response to COVID-19 is a dynamic process; however, this makes it difficult to draw conclusions about causality of our findings. Expansion of our study cohort and longitudinal sample analyses are ongoing. Third, the comparison between COVID-19 and MAS plasma cytokine levels is not patient-matched, though this is intrinsic to MAS being a predominantly paediatric condition. Using linear regression analysis to assess influence of demographic variables on our key cytokine findings (based on Del Valle et al.[52]), significant differences for CXCL8, IFN-γ and VEGF remained apparent (see Suppl. Table 3). Our study was not powered, however, to adjust for potential confounders using robust (multivariable) regression analysis. Also, although the experimental procedures to obtain cytokine levels in both cohorts were identical, plasma samples of MAS patients were stored for a longer period of time which might minimally impact results. Our findings on cytokine profiles, however, find resonance in the immune cell populations' abundance and functionality we extensively studied. Fourth, most of our analyses were performed on PBMCs, which not always reflect the ongoing disease processes at the site of infection. Nevertheless, these findings were complemented with plasma cytokine and NET biomarker measurements. Moreover, myeloid-driven inflammation was also evident from scRNA-seq data of COVID-19 BAL fluid, thereby expanding our observations to the local inflammatory compartment at the site of infection[61]. Lastly, the scope of our cytokine measurements and functional analyses of innate immune cells was narrow. For example, we cannot comment on the (direct) contribution of myeloid phagocytosis or respiratory burst-activity of neutrophils, nor on the relative contribution of different neutrophil-modulating molecules (e.g. C5a, CXCL8 isoforms)[62,63]. We do want to point out, however, that our focus on NET formation as one end-product of neutrophil hyperactivation is robust, as it is supported by our computation biology analysis evidencing neutrophil contribution to immunopathology in critical disease state, the link between NET formation and thrombosis[64] and evidence of NET formation in critical influenza immunopathology[65].

Based on our data, we propose that increased pro-inflammatory cytokine signalling parallel to a defective type II IFN response is a key mediator of critical COVID-19 pathophysiology; thereby making (a combination of) specific anti-cytokine monoclonal antibodies (e.g. anti-IL-6) or broad-activity

immunomodulatory drugs (e.g. colchicine, azithromycin) interesting candidates for interventional clinical trials. Moreover, systemic hyperinflammation might be prevented by targeting neutrophils or NETs (e.g. DNase therapy) in the lung (early in the disease course) to prevent clinical deterioration. The demonstrated efficacy of dexamethasone[66], whose mechanism of action is poorly understood but likely dampens both the dysregulated immune response and neutrophil activation seen in our study, supports further investigation along these avenues.

## Methods

**Patient cohort, sampling and data collection.** In this prospective single-centre study, adult COVID-19 patients were recruited at the COVID-19 hospitalisation wards of our tertiary care centre in Leuven (Belgium) between March 27 and April 17 2020. COVID-19 was defined as a positive qRT-PCR on respiratory sample and/or CT imaging compatible with SARS-CoV-2 disease. Patients with (i) active haematological malignancy; (ii) active infectious/inflammatory conditions other than COVID-19; (iii) calcineurin-inhibitor treatment, or (iv) patients or legal representatives unable or unwilling to give informed consent were excluded. The control population consisted of (i) 10 healthy controls recruited among hospital staff (negative COVID-19 serology); (ii) a historical cohort of 10 patients with macrophage activation syndrome (MAS) and (iii) 11 patients with non-COVID pneumonia.

COVID-19 patients were stratified by clinical status at the time of study sampling, in particular, two groups were made: mild-moderate group (either receiving no respiratory support or oxygen via nasal cannula) and critical condition group (receiving high flow oxygen support or mechanical ventilation). Blood samples from all patients were collected at the earliest possible timepoint after admission, as per study protocol. EDTA, heparin and citrate tubes were collected. Separation of plasma and PBMCs from EDTA tubes was performed using a lymphocyte separation medium (LSM, MP Biomedicals). PBMCs were frozen in 10% dimethyl sulfoxide (Sigma) and stored in liquid nitrogen for a maximum of 3 weeks, until further processing. Plasma was kept at −80 °C until processing. All patient samples were aliquoted at collection for subsequent single analytical purposes. If bronchoscopy with BAL was performed as part of the standard of medical care, a dedicated aliquot of this sample was collected and freshly processed for single-cell RNA sequencing.

Demographic, clinical, laboratory, radiologic, treatment and outcome data from patient electronic medical records (KWS v.3.3.0) were obtained through a standardized research form in Research Electronic Data Capture Software (REDCAP v.10.6.13, Vanderbilt University). Outcome data were evaluated until May 4 2020. All study procedures were in accordance with the Declaration of Helsinki and approved by the Ethics Committee of the University Hospitals Leuven. Informed consent was obtained from all individuals or their legal guardians. Research was performed as part of the COntAGIouS observational clinical trial: https://clinicaltrials.gov/ct2/show/NCT04327570.

**Chemokine and cytokine assays.** Chemokine and cytokine levels in plasma were assessed by Meso Scale Discovery using the V-plex human cytokine 30-plex kit, complemented with Human IL-1RA (V-plex), human IL-18 (U-plex) and Human CXCL9 (R-plex) kits.

**Computation systems biology.** Unbiased computational systems biology-driven modelling was performed to predict cell types responsible for COVID-19 cytokine release, using our cytokine/chemokine plasma data as input and human immune cell-type expression profiles derived from 4639 human immune cell samples assembled from 191 independently published studies. We first calculated the fold change (FC) between COVID-19 patient subgroups (mild-moderate and severe) and healthy controls for screening-derived cytokine/chemokine values. These FC values were log2 transformed and, per COVID-19 subgroup, only those cytokines/chemokines were selected that had final $log_2FC > 1$; a validated threshold[67]. These target genes were then entered into the Immuno-Navigator computational pipeline to create correlation networks per human immune cell-type gene-expression profiles. Briefly, we created co-expression networks for specific genes, wherein the genes were linked based on Pearson correlation coefficient (PCC) thresholds for creating edges. Thicker edges indicated (statistical) significance of the PCC threshold; the default Immuno-Navigator settings were used per cell. These statistical thresholds, based on primarily false discovery rates, per reference immune cell type are: B cells (edge correlation threshold = 0.4; significance correlation threshold = 0.47), CD4 T cells (0.4; 0.4), CD8 T cells (0.4; 0.49), cDCs (0.4; 0.46), macrophages (0.4; 0.43), neutrophils (0.4; 0.54), NK cells (0.4; 0.45) and pDCs (0.4; 1). To create similarity matrix analyses between cytokine/chemokine data and mass cytometry-derived peripheral immune cell enrichments per COVID-19 patient subgroup (i.e. mild-moderate vs. severe), we utilized the Morpheus software (version 1, https://software.broadinstitute.org/morpheus and https://github.com/cmap/morpheus.js) whilst considering Pearson correlation metric for correlation matrix creation and one minus Pearson correlation metric for hierarchical

clustering. Values were considered for only those COVID-19 patients who had matched analyses for both cytokine/chemokine screening as well as mass cytometry. For network analyses integrating the Gene Ontology (GO) terms specific for immunology-related biological processes, we entered the specified genes into the GOnet computational pipeline (Ontology version: 2019-07-01 and Human annotation version: 2019-07-01)[68]. Within the GOnet, the input human genes were computed for GO biological process term annotation based on predefined GO slim subset for immunology (experimental; process only) and represented via the Euler force-directed (physics simulation) layout (wherein gene-unconnected terms were hidden). This analysis reconstructs relationship between genes and GO terms thereby giving a better idea of the functional immunological impact of specific input genes.

**Mass cytometry**

*Sample processing and staining procedure.* Whole-blood (WB) samples were collected into Lithium heparin tubes and processed for mass cytometry staining within 2–4 h of isolation. WB was stained with the Maxpar Direct Immune Profiling Assay (DIPA) kit from Fluidigm© by following the workflow outlined for whole-blood staining. The last step of the protocol was performed overnight at 4 °C. Alternatively, samples that could not be acquired on the instrument the next day, were cryopreserved in the same solution at −80 °C. The cryopreservation technique was validated in triplicate by dividing aliquots of donor samples stained on the same day and comparing cell viability and immune profiles between fresh and cryopreserved samples. Batch effects were evaluated by daily running a reference sample derived from an aliquot of the same healthy donor over the period of the study.

*Data acquisition.* Cells stained for mass cytometry were acquired the day after staining or within 1 week of cryopreservation. For CyTOF acquisition, cells were pelleted in Milli-Q water on the day of acquisition and transferred to the KU Leuven Flow and Mass Cytometry Facility to be acquired on a Helios mass cytometer (Fluidigm©). Cells were resuspended into a 1 million/ml concentration with Maxpar Cell Acquisition Solution containing EQ beads diluted at 1:10. Samples were filtered directly prior to acquisition through 35 μm cell strainer cap tubes. Cells were acquired at a rate of 250–300 events per second. CyTOF software version 6.7.1016 and the Maxpar Direct Immune Profiling Assay.tem template were used to acquire and normalize data from the stained samples.

*Data analysis (Suppl. Fig. 5).* Normalized .fcs files were transferred to the Maxpar Pathsetter™ software (version 2.0.45) for QC (including bead removal and high-quality singlet selection). In-depth data analysis was subsequently done using three parallel strategies. First, the built-in immunoprofiling tools of the Maxpar Pathsetter™ software were used to analyse the overall immune cell population in a highly standardized and automated way. Second, we used 123 cleaned .fcs files, including 8 healthy controls and 115 COVID-19 patients, through various stages of the disease. Samples were manually gated for live single cells, and samples with fewer than 50,000 cells were discarded. Samples were then preprocessed: margin events were filtered out, data was transformed with an arcsinh transformation with cofactor 5 and the PeacoQC algorithm (v.0.99.30) was applied to remove any unstable signal regions during the measurement[69]. A principal component analysis of the 25, 50 and 75 percent quantiles of the marker values marked 4 additional files as outliers, which were not taken along further in the analysis. On this cleaned data a first FlowSOM model (v.2.1.8) was trained[70], using a random selection of cells for all samples, resulting in 3,000,093 cells to train on. The clustering made use of 11 markers (CD45, CD66b, CD3, CD4, CD8a, TCRgd, NCAM, CD11c, CD19, CD14 and CD20), mapped the data onto a 10 by 10 SOM grid and resulted in 30 meta-clusters. Twenty-two meta-clusters were selected as having CD66b values lower than 2 or CD45 values higher than 4, corresponding to non-granulocytes, while 8 meta-clusters were labelled as granulocytes. The full files were mapped onto this model, and for each of them new fcs files were generated corresponding to the two subsets of cells. A second FlowSOM model was built including only the non-granulocyte cells (or only granulocyte cells), again using only a subset of 2,949,946 (granulocyte: 3,000,093) cells for training mapped onto a 10 by 10 SOM grid, this time using 33 markers (CD45, CCR6, IL-3R, CD19, CD4, CD8a, CD11c, CD16, CD45RO, CD45RA, CD161, CCR4, IL-2Ra, CD27, CD57, CXCR3, CXCR5, CD28, CD38, CD69, NCAM, TCRgd, CD163, CD294, CCR7, CD14, NKG2A, CD3, CD20, CD66b, HLA-DR, IgD and IL-7Ra). To be able to identify small populations, no meta-clustering was applied on these second models, and the 100 clusters were manually annotated by 3 independent experts according to their mean fluorescence intensity (MFI) values. In the non-granulocyte model, 6 clusters were manually identified as still being mixtures of different cell types, and split into 2 or 3 clusters, resulting in 107 non-granulocyte clusters and 100 granulocyte clusters. These were themselves clustered by hierarchical clustering with complete linkage. Finally, 54 samples (8 healthy controls, 32 selected patient samples labelled as mild-moderate and 14 selected samples labelled as critical disease) were mapped onto these models to identify their immune profiles. Because different clustering methods can generate different results, we applied 3 clustering methods (Pheno-Graph, FlowSom, and KMeans[71]) and followed a wisdom-of-the-crowds type of approach to identify the different immune cell populations. Briefly, MFIs were asinh transformed and each marker was normalized in the [0–5] range using q99

normalization. A randomly sampled subset of 1e5 cells was used for the initial clustering using a subset of 12 markers (CD19, CD3, CD14, CD11c, CD4, CD45, CD20, CD8, CD56, TCRgd, CD66b, and CD294) and the three clustering methods abovementioned. Clusters were manually annotated to known cell phenotypes by two independent experts (FMB, FDS). To perform a comparative assessment, we correlated the various approaches. Overall, all clustering algorithms concurred very well. Final annotations were defined by a consensus-based approach where only those cells that agreed over at least 2 algorithms were pertained. As such, we observed that >92% of the cells showed agreement across all clustering methods, ~7% of the cells showed agreement in 2 of 3 clustering methods, and ~1% of the cells were discarded due to annotation disagreement. Cell phenotypes were represented by an expression fingerprint summarizing the average expression of all its cells for each of the 12 markers. These fingerprints were used to make predictions on the whole population of cells. Each identified main population was further clustered using the PhenoGraph clustering method and the whole set of markers. Cluster annotation, fingerprint construction, and population prediction was performed in the same way as the first iteration. Once annotated in all 3 approaches, cluster percentages, MFI values and between-cluster-ratios and sums (according to the merge-hierarchy of the hierarchical clustering) were compared between the groups with Wilcoxon rank tests and fold changes of the group medians. In addition, a UMAP dimensionality reduction was computed on a subset of 50,000 cells from these samples (using the uwot R package with default parameters). The code used to generate these results is available at https://github.com/saeyslab/CYTOF_covid19_study. Analyses were performed in R version 4.0.

**Flow cytometry**

*Sample processing and staining procedure.* Frozen PBMCs were thawed, plated and incubated for 4 h with complete RPMI containing phorbol myristate acetate (PMA 50 ng/mL), ionomycin (500 ng/mL) and Brefeldin A (8 µg/mL; all Tocris Bioscience, Bristol, UK) at 37 °C with 5% CO$_2$. Cells were then washed twice with PBS (Fisher Scientific, Hampton, NH, USA) and stained with live/dead marker (fixable viability dye eFluor780; eBioscience, San Diego, CA, USA) and fluorochrome-conjugated antibodies against surface markers: anti-CD14 (TuK4, 1:200, MHCD1418), anti-CCR7 (G043H7, 1:50, 25-1979) (eBioscience); anti-CD3 (REA613, 1:50, 130-113) (Miltenyi Biotec, Bergisch Gladbach, Germany); anti-CD4 (SK3, 1:50, 564651), anti-CD8 (SK1, 1:200, 564912), anti-PD-1 (EH12.1, 1:25, customed Ab), anti-CD45RA (HI100, 1:50, 612926) (all from BDBiosciences, San Jose, CA, USA); and anti-CD25 (BC96, 1:25, 302636), anti-HLA-DR (L243, 1:40, 307638), anti-CD40L (24–31, 1:25, 310842), anti-4-1BB (4B4-1, 1:25, 309822), anti-CD19 (HIB19, 1:20, 302242) (all from BioLegend, San Diego, CA, USA). Cells were fixed with 2% Formaldehyde (VWR chemicals, Radnor, Pennsylvania, PA, USA) and then permeabilised with eBioscience permeabilisation buffer according to manufacturer's instructions. Cells were stained overnight at 4 °C with anti-IFNγ (4 S.B3, 1:75, 564620), anti-IL-6 (MQ2-13A5, 1:75, 563543), anti-IL17a (N49-653, 1:50, 565163), anti-RORyt (Q21-559, 1:75, 563081), anti-IL-2 (MQ1-17H12, 1:75, customed Ab), anti-IL-10 (JES3-9D7, 1:50, 564051), anti-T-bet (4B10, 1:50, customed Ab), anti-CTLA-4 (BNI3, 1:75, 555854), anti-GATA3 (L50-823, 1:50, 565448) (all from BDBiosciences); anti-IL-4 (MP4-25D2, 1:75, 500826), anti-TNFα (Mab11, 1:50, 502915), anti-FOXP3 (206D, 1:50, 320114) (all from BioLegend).

*Data acquisition and analysis (Suppl. Fig. 6c–e).* Data were acquired on a BD Symphony, up to $5 \times 10^5$ cells were acquired for each sample. Classical manual gating strategy was applied in FlowJo (version 10.6.1) and cell subsets were defined by well-described surface markers. Compensation was performed using Autospill[72]. The complete set of FCS files used for the COVID-19 cytokine immune phenotyping has been deposited on FlowRepository and annotated in accordance with the MIFlowCyt standard. These files may be downloaded for further analysis from https://flowrepository.org/experiments/2713.

**Single-cell RNA sequencing.** Single-cell RNA sequencing was performed on 13 'mild-moderate' and 10 'critical' disease PBMC samples as well as 11 fresh BAL samples, sequencing 60675, 22849 and 26605 cells, respectively (Suppl. Figs. 4a and 8a). Single-cell suspensions were converted to barcoded scRNA-seq libraries by using the Chromium Single Cell 5′ library and Gel Bead & Multiplex Kit from 10x Genomics. Libraries were sequenced on an Illumina NovaSeq6000, and mapped to the human genome GRCh38 using CellRanger (10x Genomics). Raw gene-expression matrices generated per sample were merged and analysed with the Seurat package (v3.1.4).

*Preparation of single-cell suspensions. BAL fluid:* Approximately 10 ml of BALF was obtained and placed on ice, with processing within 3 h in a BSL-3 laboratory. BAL fluid was centrifuged and the supernatant was frozen at −80 °C for further experiments. The cellular fraction was resuspended in ice-cold PBS and samples were filtered using 40 µm nylon mesh (ThermoFisher Scientific). Following centrifugation, the supernatant was decanted and discarded, and the cell pellet was resuspended in red blood cell lysis buffer. Following a 5-min incubation at room temperature, samples were centrifuged and resuspended in PBS containing

UltraPure BSA (AM2616, ThermoFisher Scientific) and filtered over Flowmi 40 µm cell strainers (VWR) using wide-bore 1 ml low-retention filter tips (Mettler-Toledo). Next, 10 µl of this cell suspension was counted using an automated cell counter to determine the concentration of live cells. The entire procedure was completed in <1.5 h.

*PBMCs:* PBMC samples were thawed, centrifuged and the resulting cellular fraction resuspended in PBS containing UltraPure BSA. This was followed by filtering and counting, according to BALF protocol. The entire procedure was completed in <1 h.

*Single-cell RNA-seq data acquisition and pre-processing.* Libraries for scRNA-seq were generated using the Chromium Single Cell 5′ library and Gel Bead & Multiplex Kit from 10x Genomics. We aimed to profile 5000 cells per library. All libraries were sequenced on Illumina NovaSeq6000 until sufficient saturation was reached. After quality control, raw sequencing reads were aligned to the human reference genome GRCh38 and processed to a matrix representing the UMI's per cell barcode per gene using CellRanger (10x Genomics, v3.1). Multiplex sequencing was performed for PBMC samples, pooling 2 donors. Data were deconvolved using SCsplit[73] and annotated using SNPs and sex chromosomes.

*Single-cell RNA analysis to determine major cell types and cell phenotypes.* Raw gene-expression matrices generated per sample were merged and analysed with the Seurat package (v3.1.4). PBMC matrices were filtered by removing cell barcodes with <401 UMIs, <201 expressed genes, >6000 expressed genes or >25% of reads mapping to mitochondrial RNA. The remaining cells were normalized and the 2000 most variable genes were selected to perform a PCA analysis after regression for confounding factors: number of UMIs, % of mitochondrial RNA, patient ID, cell cycle (S and G2M phase), hypoxia, stress and interferon score. PCs ($n = 11$) covering the highest variance in the dataset were selected based on an elbow plot. Clusters were calculated by the FindClusters function with a resolution between 0.1 and 1.5, and visualised using the UMAP dimensional reduction method, a resolution of 1.0 was selected since all known cell types were identified as a cluster at this given resolution. Clusters were annotated based on the expression of marker genes (Suppl. Fig. 4a–c).

To preserve neutrophils, which are transcriptionally less active (lower transcripts and genes detected), we slightly modified the filtering parameters for BAL fluid samples and removed cell barcodes with <301 UMIs, <151 expressed genes or >20% of reads mapping to mitochondrial RNA. Similar PCA and graph-based clustering approach resulted in some highly patient-specific clusters, which prompted us to perform a data integration using CCA in Seurat(v3) package between patients to reduce the patient-specific bias. After data integration, mitochondrial, cell cycle, hypoxia, stress and interferon response genes were removed from the variable genes used for downstream PCA, graph-based clustering and marker gene-based cluster annotation (Suppl. Fig. 8a–c).

Neutrophils in BAL fluid samples and T cells and monocytes in PBMCs were further subclustered using the same strategy. Neutrophil subclustering revealed one low quality and one doublet cluster (266 cells in total), which were removed for further analyses, identifying an 'active' and 'resting' neutrophil population. Doublet clusters expressed marker genes from other cell lineages, and had a higher than expected doublets rate, as predicted by the artificial k-nearest neighbours algorithm implemented in DoubletFinder (v2).

**Net analysis.** Platelet-poor plasma was prepared from freshly drawn citrate blood tubes centrifuged at $400 \times g$ for 7 min at room temperature, followed by a second centrifugation of the supernatant at $3000 \times g$ for 7 min. Plasma was collected and stored at −80 °C until analysis.

Plasma was diluted 1:100 for analysis of myeloperoxidase antigen levels using the LEGEND MAX$^{TM}$ Human Myeloperoxidase ELISA kit (Biolegend) according to manufacturer instructions, and diluted 1:4 for analysis of NET biomarkers (MPO-DNA complexes and citrullinated histone H3). MPO-DNA complexes were measured using an in-house ELISA modified from the Cell Death Detection ELISA (Roche). A 96-well Nunc immunoassay plate (MediSORP, ThermoFisher) was coated overnight with polyclonal anti-myeloperoxidase antibody (1:1000 dilution, ThermoFisher PA5-16672) in 0.05 M sodium carbonate/sodium bicarbonate buffer (pH 9.6). After 4 washes with PBS containing 0.05% Tween-20, wells were blocked with 3% bovine serum albumin. Samples were diluted in assay buffer (0.3% BSA) and incubated for 2 h in duplicate wells. Following extensive washing, wells were incubated with mouse anti-DNA monoclonal antibody conjugated with peroxidase from the Roche Cell Death Detection ELISA, washed, and detected with ready-to-use TMB substrate (Life Technologies, 2023). The reaction was stopped with 1 N hydrochloric acid and the plate read at 450 nM with 630 nM background subtraction using a Biotek Gen5 microplate reader. Values were normalized to a plasma pool from 10 healthy volunteers as multiple plates were needed to perform the full analysis. An in vitro prepared positive-control standard using the MPO standard from the LEGEND MAX$^{TM}$ Human Myeloperoxidase ELISA kit incubated with native human nucleosomes (Merck Millipore, 14-1057) confirmed specificity of the assay and provided an estimated detection range from 49.79 to 183.3 ng/ml. Citrullinated histone H3 levels were measured according to manufacturer instructions with the Citrullinated Histone H3 [clone 11D3] ELISA kit from Cayman Chemicals.

**Chest computed tomography and score assessment**. All CT scans were performed using a Siemens SOMATOM Definition Flash, dedicated to the COVID-19 emergency department of our institution.

Chest radiologists performed qualitative and quantitative evaluations of lung parenchyma opacities on CT scan. A CT score was assigned by converting percentage of lung parenchyma opacity for each lobe into a 5-points Likert scale: a score of 0 for 0% lung opacity (LO), 1 for 1% to <5% LO, 2 for 5–25% LO, 3 for 26–50% LO, 4 for 51–75% LO, and 5 for 76–100 LO. The total CT score is the sum of the individual lobar scores and can range from 0 (no area with increase in lung opacity) to 25 (all five lobes show more than 75% increase in lung opacity).

**Quantification and statistical analysis**. Descriptive statistics are presented as median [interquartile range; IQR] and $n$ (%) for continuous and categorical variables, respectively. The Mann–Whitney U test and Kruskal–Wallis test with Dunn's correction for multiple comparisons were used to compare differences in continuous data between groups as appropriate. Pearson's $\chi^2$ or Fisher's Exact test was used to compare differences in non-ordered categorical data between patient groups. Available case analysis was implemented to address data missingness where appropriate. Correlation analyses were tested by simple linear regression. Statistical analyses were performed using R (version 3.6.3, R Foundation for Statistical Computing, R Core Team, Vienna, Austria) in the RStudio integrated development environment (version 2.2.1; RStudio, Inc., Boston, MA, USA) and Graphpad Prism version 8.4.2. Statistical analyses were performed with a two-sided alternative hypothesis at the 5% significance level.

**Reporting summary**. Further information on research design is available in the Nature Research Reporting Summary linked to this article.

## Data availability

The data supporting the findings regarding cytokine and neutrophil activation biomarker experiments, are available within the paper and its supplementary information files. The mass cytometry raw data files are publicly available on FlowRepository with Repository ID FR-FCM-Z2MW (https://flowrepository.org/experiments/2770). Regarding flow cytometry data, the complete set of FCS files has been deposited on FlowRepository with Repository ID FR-FCM-Z2KP and may be downloaded for further analysis from https://flowrepository.org/experiments/2713. Raw sequencing reads of the scRNA-seq experiments generated for this study have been deposited in the EGA European Genome-Phenome Archive database (EGAS00001005039 for PBMC data accessible at: https://ega-archive.org/studies/EGAS00001005039; EGAS00001004717 for BAL fluid data accessible at: https://ega-archive.org/studies/EGAS00001004717). Based on SCope, which is an interactive web server for scRNA-seq data visualisation, a download of the scRNA-seq read count matrices is also available at http://covid19.lambrechtslab.org/. Publicly available data that were used to support this study are available from Gene Expression Omnibus GSE150728. Source data are provided with this paper.

## Code availability

Data processing steps are described in the "Methods" section. R scripts to analyse the mass cytometry data can be found at https://github.com/saeyslab/CYTOF_covid19_study.

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

## Acknowledgements

We acknowledge Tiene Callewaert, Sofie Coenen, Cato Jacobs, Silke Janssens, Rita Merckx, Tania Mitera, Noëmie Pörtner, Kim Rottiers, Evi Smeyers and Lotte Vanbrabant for sample processing and data collection; Maylorie 't Lam, Kaat Haesendonck and Robin Justé for sample collection; Reena Chinnaraj, Tatjana Verbeke, Marleen Derweduwe, Annelies Claeys for technical support for the CyTOF experiments; Thomas Van Brussel and Rogier Schepers for technical support for scRNA-seq experiments; Elisabeth Heylen for coordinating experiments in the BSL-3 laboratory. This work was funded by KU Leuven (internal fund and grant C14/17/084 and C16/17/010), VIB (Grand Challenge project), UZ Leuven (KOOR project), FWO grant I007418N, the Rega Foundation (research expert fellowship to G.M.) and 'het Leuvens Kankerinstituut'. The resources and services used for scRNA-seq analyses were provided by the VSC (Flemish Super-computer Center), funded by the Research Foundation—Flanders (FWO) and the Flemish Government. E.D. is a postdoctoral research fellow of the Research Foundation—Flanders (FWO), Belgium (grant number 12X9420N). J.G. holds a postdoctoral research fellowship granted by the clinical research and education council of the University Hospitals Leuven. Y.S. received funding from the Flemish Government (AI Research Programme). L.V. is supported by an FWO PhD fellowship (grant number 11E9819N). P.V.M. is supported by an FWO PhD fellowship (grant number 1S66020N). S.V.G. is an ISAC Marylou Ingram Scholar and supported by an FWO postdoctoral research grant (Research Foundation—Flanders). E.W. is supported by Stichting tegen Kanker (Mandate for basic & clinical oncology research). J.W. is supported by an FWO Fundamental Clinical Mandate (1833317 N).

## Author contributions

L.V., P.V.M., Y.V.H., F.D.S., S.H.B., K.M., A.L., P.P., AD.G., P.Ma., C.W., D.L., E.W. and J.W. designed the experiments, developed the methodology, analysed and interpreted data and wrote the manuscript. C.D., J.G., G.H., N.L., P.Me., J.O., E.P., S.R., D.T., A.W. and J.Y. performed sample collection. A.A., FM.B., M.C., D.D., W.D.W., A.E., J.F., M.G., S.J., K.L., J.N., D.P., PA.P., K.Q., J.R., Y.S., J.S., T.V.B., J.V., S.V.G., L.C.VP. and B.W. performed experiments and acquired data. I.A., B.B., E.D., T.M., J.Q., K.T., S.T. and R.V. provided technical support and supported data analysis and interpretation. E.W. and J.W. supervised the study and were responsible for coordination and strategy. All authors have approved the final manuscript for publication.

## Competing interests

The authors declare no competing interests.

## Additional information

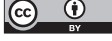

[1]Laboratory of Clinical Bacteriology and Mycology, Department of Microbiology, Immunology and Transplantation, KU Leuven, Leuven, Belgium. [2]Laboratory of Translational Genetics, Department of Human Genetics, VIB-KU Leuven, Leuven, Belgium. [3]Laboratory of Experimental Oncology, Department of Oncology, KU Leuven, Leuven, Belgium. [4]Laboratory for Precision Cancer Medicine, Translational Cell and Tissue Research, Department of Imaging & Pathology, KU Leuven, Leuven, Belgium. [5]Adaptive Immunology, Department of Microbiology, Immunology and Transplantation, KU Leuven, Leuven, Belgium. [6]Centre for Molecular and Vascular Biology, Department of Cardiovascular Sciences, KU Leuven, Leuven, Belgium. [7]Translational Cell & Tissue Research, Department of Imaging & Pathology, KU Leuven, Leuven, Belgium. [8]Laboratory of Intensive Care Medicine, Department of Cellular and Molecular Medicine, KU Leuven, Leuven, Belgium. [9]Radiology, Department of Imaging & Pathology, KU Leuven, Leuven, Belgium. [10]Laboratory of Respiratory Diseases and Thoracic Surgery (BREATHE), Department of Chronic Diseases and Metabolism, KU Leuven, Leuven, Belgium. [11]Clinical Pharmacology and Pharmacotherapy, Department of Pharmaceutical and Pharmacological Sciences, KU Leuven, Leuven, Belgium. [12]Department of Applied Mathematics, Computer Science and Statistics, VIB-UGent Center for Inflammation Research, VIB-UGent, Gent, Belgium. [13]Laboratory of Immunobiology, Department of Microbiology, Immunology and Transplantation, Rega Institute, KU Leuven, Leuven, Belgium. [14]Laboratory of Molecular Immunology, Department of Microbiology, Immunology and Transplantation, Rega Institute, KU Leuven, Leuven, Belgium. [15]Laboratory of Virology and Chemotherapy, Department of Microbiology, Immunology and Transplantation, Rega Institute, KU Leuven, B Leuven, Belgium. [16]Laboratory of Lymphocyte Signalling and Development, The Babraham Institute, Babraham Research Campus, Cambridge, UK. [17]Department of Pneumology, University Hospitals Leuven, Leuven, Belgium. [18]Laboratory for Clinical Infectious and Inflammatory Disorders, Department of Microbiology, Immunology and Transplantation, KU Leuven, Leuven, Belgium. [19]Department of Internal Medicine, University Hospitals Leuven, Leuven, Belgium. [20]KU Leuven Flow & Mass Cytometry Facility, KU Leuven, Leuven, Belgium. [21]Laboratory of Molecular Bacteriology (Rega Institute), Department of Microbiology, Immunology and Transplantation, KU Leuven, and VIB Center for Microbiology, Leuven, Belgium. [22]Anesthesiology and Algology, Department of Cardiovascular Sciences, KU Leuven, Leuven, Belgium. [23]Laboratory for Cell Stress & Immunity (CSI), Department of Cellular and Molecular Medicine (CMM), KU Leuven, Leuven, Belgium. [24]Molecular Digestive Oncology, Department of Oncology, KU Leuven, Leuven, Belgium. [25]Centre of Microbial and Plant Genetics, Department of Microbial and Molecular Systems (M2S), KU Leuven, Leuven, Belgium. [26]These authors contributed equally: L. Vanderbeke, P. Van Mol, Y. Van Herck, F. De Smet, S. Humblet-Baron, K. Martinod. [27]These authors jointly supervised this work: A. D. Garg, P. Matthys, C. Wouters, D. Lambrechts, E. Wauters, J. Wauters. ✉email: els.wauters@kuleuven.be

