## [Peer Review File · Nature Communications]

REVIEWER COMMENTS

Reviewer #1 (Remarks to the Author):

In this manuscript titled "Monocyte-driven atypical cytokine storm and aberrant neutrophil activation as key mediators of COVID-19 disease severity" Vanderbeke L et al., examine the basis for immunopathological response to SARS-CoV-2 infection. The authors first compare inflammatory response profile of COVID19 patients with that of Macrophage Activation Syndrome (MAS) patients. Based on IFN-g and its ISG expression profile, the authors conclude that COVID19 patients likely do not experience MAS, and that the cytokine profile is driven largely by classical inflammatory monocytes and dysregulated neutrophil response unique to COVID19. The latter conclusions are derived from comparison of immune response between mild to moderate and severe/critical COVID19 patients. The authors perform very compelling computational systems modeling of cytokine profiles followed by an in-depth immune cell profiling using mass cytometry and scRNA-seq on PBMCs isolated from individuals with mild and severe COVID19. These analyses show strong association between inflammatory cytokine expression and classical monocytes. The experiments are well planned and the manuscript is well written. Data largely support the conclusions reached. The data presented are very convincing and will fill critical knowledge gap in our understanding of the COVID19 pathogenesis.

This reviewer has a few comments that need to be addressed. The comments are listed below.

- 1) Comparison of MAS and COVID: Based on the IFN-g/ISG cytokine profile the authors conclude that COVID19 is different from MAS. However, the authors compare cytokine profile between total COVID19 and MAS patients. Considering MAS is more severe disease, an accurate comparison and thus correct conclusion can only be made by comparing SEVERE COVID19 patients with those of MAS. Therefore, this reviewer suggests authors to include data comparing MAS vs severe COVID19.
- 2) Figure 1B/line 139-140: The authors conclude that compromised IFN γ levels in critically ill COVID19 patients is striking. However, Fig 1b shows no difference in IFN-g levels between mild/moderate vs critically ill patients. Please correct and modify the conclusion accordingly.
- 3) Figure 4: The basis for authors conclusion that that neutrophils are dysregulated and are key to immunopathology is unclear. Please clarify or better explain correlation clusters of different cytokines that were used to reach these conclusions.
- 4) While neutrophils function is dysregulated and are highly inflammatory, the neutrophil chemoattractant CXCL8 levels are similar in mild vs critically ill COVID19 patients. please explain.
- 5) Myeloid cells are not infected by SARS-coV-2, yet they are highly pro-inflammatory. Please explain the basis for this in the discussion section.
- 6) Another major caveat, similar to a large number of recent literature is that this study relies heavily on the results obtained from PBMC. It is important to acknowledge this fact.
- 7)

Reviewer #2 (Remarks to the Author):

Thank you for the opportunity to review Vanderbeke et. al. "Monocyte-driven atypical cytokine storm and aberrant neutrophil activation as key mediators of COVID-19 disease severity". In this study, the authors employed a systems immunology approach to study differences in plasma inflammatory proteins and immune cell properties in patients with mild/moderate and critical COVID-19 disease, as well as patients with non-COVID19 Macrophage Activation Syndrome (MAS). The authors use a combination of analytical methods, including multiplex quantification of plasma protein

concentration, analysis of immune cell distribution and function with mass cytometry and fluorescence flow cytometry, single-cell RNAseq of immune cell in PBMC, and bronchoalveolar lavage, and analysis of NET formation. The authors also use an extensive array of bioinformatic methods to parse through the complex data thus generated, which yields interesting new findings regarding the role of monocyte and neutrophils in the pathobiology of COVID19. From a technical perspective, the proteomic, mass cytometry, and scRNAseq data generated is impressive and appropriate measures are utilized to ensure data quality. My comments/concerns specific to the systems immunology approach are:

1)The authors generated a complex and multimodal dataset that provides a rich characterization of peripheral and local host immune response. However, it is unclear what the primary outcome of the study is, or what primary comparisons drive the study design. For example, in Fig. 1a the focus is on MAS vs COVID19 patients, while Fig 1b is on COVID19 patient severity. Similarly, Fig 2a shows a comparison of COVID19 disease severity, while it is unclear what patient groups are being compared for Fig 2b. This is a major concern as additional group comparison increases the number of hypotheses tested significantly.

2)The statistical methods for evaluating patient group differences are not always obvious. While the authors describe in detail the computational approaches used for data processing and feature identification (for instance, multiple methods for immune cell identification and enumeration), insufficient information is provided for the reader to understand how patient cohorts are being compared to each other. For example, in Suppl. Fig 3, the statistical basis for the identification of differences in correlation networks between patient groups is unclear.

3) Similarly, for the scRNAseq data presented in Fig 2, it is not obvious whether groups of patients are being compared to one another, or whether the analysis focuses on differences between cell subsets. If it is the latter and the goal is to determine the cellular source of cytokines/chemokines increased in patients with COVID-19, it would be useful to know whether scRNAseq data correlate with plasma cytokine concentration for the relevant cytokines.

4)The data provided in Fig. 4 (similarity matrix) provides only qualitative observations and a quantitative evaluation of between-group differences would be helpful.

5)With respect to the mass cytometry data, three approaches are utilized for cell cluster identification. While multiple approaches can be useful in evaluating the dependence of a finding on a particular cell-type identification method, the authors – to my knowledge – did not provide this information. A justification for the three approaches described in the methods and a comparative assessment would be useful.

6)Some of the observed differences between patient cohorts could be driven by clinical or demographic variables other than the studied diseases. This is particularly important for the data presented in Fig. 1 where samples from COVID19 patients are compared to historical samples. How did the author ensure that observed differences are not driven by confounding technical (sampling time), clinical or demographic variables?

Reviewer #3 (Remarks to the Author):

This manuscript presents a comprehensive analysis of immune correlates of moderate and severe SARS-CoV-2 infections. The authors use state-of-the-art immunology, RNAseq and computational modelling to establish beyond doubt that COVID-19 patients show a “cytokine storm” that is phenotypically distinct from the canonical macrophage activation syndrome. Specifically, IFN- γ levels are much lower than expected and IFN- γ responses are much lower than in a “conventional” cytokine storm. This is driven in COVID-19 cases by monocytes and results in depressed antigen presentation and lowered Th1 and CD8+ cell levels. There is also evidence of a huge increase in neutrophil activation at sites of infection. Taken together these observations help explain the pathology of COVID-19 infection.

This is a very important manuscript that will help us understand COVID-19 pathology and may help with treatment regimes. It will also encourage the immunology community to think more broadly about using the dreadful phrase “cytokine storm” – it would appear that different pathogens can induce distinct types of response, and whilst perhaps not surprising, this should be more widely known. A clear demonstration using such a contemporary problem is to be welcomed.

I can't think of anything else they could have done to establish their case more clearly and this will undoubtedly be a work of reference in the field. It's not really a criticism but the manuscript is very descriptive; in trying to be as thorough (and non-committal) as possible it is a very “dry” read and will be impenetrable to anyone other than professional immunologists. For example, how many readers will know what “non-classical monocytes” are, and more importantly appreciate that these may play a significant role in antiviral defence and so the relative loss of these cells is probably very important in the disease? More narrative throughout the text would help non-experts to navigate. Similarly, although many will be happy with the concept that COVID-19 is predominantly a disease of immunopathology, as the authors go to great lengths to point out, this is an atypical (or at least uncommon) form of immunopathology for a respiratory infection; how does this square with the opening line of the summary – “host immune response to SARS-CoV-2, more so than viral factors, determines COVID-19 severity”? Surely because the immune response to SARS-CoV-2 is distinct the virus must be in some way responsible for this, even if the pathological outcome is mediated by host immunity? The discussion would be improved by trying to contextualise the immunology in terms of the virus.

Steve Goodbourn

Monocyte-driven atypical cytokine storm and aberrant neutrophil activation as key mediators of COVID-19 disease severity

Rebuttal Letter

We thank all reviewers for their constructive and supportive comments, that have greatly helped to position our findings within the rapidly evolving COVID-19 scientific and clinical framework. We have made considerable efforts to respond to each of them in a point-by-point manner in this letter, and more narratively in the revised manuscript (with changes highlighted in yellow).

REVIEWER I COMMENTS

In this manuscript titled "Monocyte-driven atypical cytokine storm and aberrant neutrophil activation as key mediators of COVID-19 disease severity" Vanderbeke L et al., examine the basis for immunopathological response to SARS-CoV-2 infection. The authors first compare inflammatory response profile of COVID-19 patients with that of Macrophage Activation Syndrome (MAS) patients. Based on IFN- γ and its ISG expression profile, the authors conclude that COVID-19 patients likely do not experience MAS, and that the cytokine profile is driven largely by classical inflammatory monocytes and dysregulated neutrophil response unique to COVID-19. The latter conclusions are derived from comparison of immune response between mild to moderate and severe/critical COVID-19 patients. The authors perform very compelling computational systems modeling of cytokine profiles followed by an in-depth immune cell profiling using mass cytometry and scRNA-seq on PBMCs isolated from individuals with mild and severe COVID-19. These analyses show strong association between inflammatory cytokine expression and classical monocytes. The experiments are well planned and the manuscript is well written. Data largely support the conclusions reached. The data presented are very convincing and will fill critical knowledge gap in our understanding of the COVID-19 pathogenesis. This reviewer has a few comments that need to be addressed. The comments are listed below.

1) Comparison of MAS and COVID: Based on the IFN- γ /ISG cytokine profile the authors conclude that COVID-19 is different from MAS. However, the authors compare cytokine profile between total COVID-19 and MAS patients. Considering MAS is more severe disease, an accurate comparison and thus correct conclusion can only be made by comparing SEVERE COVID-19 patients with those of MAS. Therefore, this reviewer suggests authors to include data comparing MAS vs severe COVID-19.

We thank the reviewer for this valid comment. The cytokine data has now been reanalyzed comparing MAS cases with COVID-19 patients in critical clinical condition. Our most important findings, namely reduced type II interferon signaling, increased CXCL8 levels and increased VEGF comparing COVID-19 to MAS remain valid when comparing critical COVID-19 to MAS. Also, the comparison of laboratory values between COVID-19 and MAS has been reanalyzed. Exact p-values are slightly adapted yet P-values significant (or non-significant) for all previously identified parameters remained, except for aspartate aminotransferase levels (no longer significantly different between MAS and COVID-19 critical disease).

These additional analyses are discussed in the main text (lines 100-133), and we have adapted the figures and tables accordingly:

- Figure 1a: Hypercytokinemia in COVID-19 as a distinct cytokine release syndrome.

Comparison of plasma levels of selected cytokines and chemokines from healthy controls (n=10), COVID-19 critical condition (n=22) and MAS patients (n=10) (a)

- Supplementary figure 1: Cytokine and chemokine plasma levels: comparison with MAS

Overview of cytokine and chemokine levels with comparison between healthy controls (n=10), COVID-19 critical condition (n=22) and MAS (n=10) patients.

- Supplementary table 1: Laboratory findings of patients infected with MAS versus COVID-19 critical condition and within COVID-19 clinical condition subgroups.

2) Figure 1B/line 139-140: The authors conclude that compromised IFN- γ levels in critically ill COVID-19 patients is striking. However, Fig 1b shows no difference in IFN- γ levels between mild/moderate vs critically ill patients. Please correct and modify the conclusion accordingly.

This is a correct interpretation of the data. We have adapted the conclusion as follows in the manuscript (line 148-150): "The compromised production of IFN- γ , a key cytokine in antigen presentation and development of adaptive immune responses, is striking in all COVID-19 patients regardless of disease severity."

3) Figure 4: The basis for authors' conclusion that neutrophils are dysregulated and are key to immunopathology is unclear. Please clarify or better explain correlation clusters of different cytokines that were used to reach these conclusions.

We thank the reviewer for pointing out that this conclusion should be better explained and refer to the data in Suppl. Fig 3b (lower figure subpanels). Herein, the GO terms that are most often enriched in mild-moderate COVID-19 patients (indicated in yellow color) mainly include, "defense response", "positive regulation of immune system", "cell motility" and "leukocyte migration"; all of which are a sign of a normal innate immune response. However, a similar analysis in critical COVID-19 patients shows enrichment of GO biological process terms similar to the ones stated above, but further combined with "regulation of programmed cell death". This suggests the possibility of increased cellular stress or cell death in neutrophils (i.e. a dysregulated response) brought about by inflammation. We have included this explanation in the manuscript (line 280-287).

Besides this correlation analysis however, our conclusion of a dysregulated neutrophil compartment was also based on these additional findings, 1) increased number of neutrophils in blood of critically ill COVID-19 patients as evidenced by mass cytometry 2) high serum levels of CXCL8 in COVID-19 patients (and not in MAS patients) 3) increased levels of neutrophil activation and NET formation markers in PBMCs of COVID-19 versus non-COVID patients (including significant differences between mild and critically ill patients) 4) the higher number of active neutrophils in BAL fluid of COVID-19 patients versus non-COVID pneumonia patients. We have carefully rephrased the revised manuscript by highlighting these observations more clearly.

4) While neutrophils' function is dysregulated and they are highly inflammatory, the neutrophil chemoattractant CXCL8 levels are similar in mild vs critically ill COVID-19 patients. Please explain.

We uncover the highly inflammatory aspect of neutrophils in COVID-19 patients, based on both plasma NETosis biomarker measurements and bronchoalveolar lavage fluid scRNA-seq experiments. We also report increased blood neutrophilia in COVID-19 patients compared to healthy controls, more so in critical than in mild-moderate condition, while plasma CXCL8 levels were similar in the latter two groups. We understand the apparent discrepancy in these findings as highlighted by the reviewer, since CXCL8 is known as the most potent chemoattractant for neutrophils in humans.

However, neutrophil migration and activation is not solely established through interaction of CXCL8 with its receptors. Other, both non-specific and neutrophil-specific, chemotactic molecules are also involved, such as leukotriene B₄, formyl peptides, complement factor C5a and CXCL1 to CXCL7¹. These were not assessed in our multiplex chemokine and cytokine assay, yet are likely to contribute to the COVID-19 inflammatory setting and thus influence neutrophil numbers in blood. In fact, it has already been shown that the anaphylatoxin C5a is increased in COVID-19, in proportion to disease severity². In addition, we did not investigate the different isoforms of CXCL8 present in plasma of COVID-19 patients, since this was not the aim of this study. As posttranslational modifications of CXCL8 are known to be responsible for more than 10 differently processed, naturally occurring isoforms with various biological activity³, their relative contribution in the different COVID-19 patient groups could provide an additional explanation of the apparent discrepancy highlighted by the reviewer.

As there is no description of a direct link between CXCL8 levels, neutrophil count and inflammatory status and severity of COVID-19 in our manuscript, we did not elaborate on this remark of the reviewer within the "Results" section of the manuscript, yet have acknowledged the above as a limitation of our study in the revised discussion (line 449-453).

5) Myeloid cells are not infected by SARS-CoV-2, yet they are highly pro-inflammatory. Please explain the basis for this in the discussion section.

Vabret et al. provide a review of hypotheses based on clinical data obtained during the MERS-CoV, SARS-CoV-1 and SARS-CoV-2 epidemics as well as on preclinical data⁴. They sketch a framework, in line with our findings, in which the myeloid immune compartment is characterized by an untimely antiviral response, linked to a heightened proinflammatory reaction.

How exactly pathogenic coronaviruses trigger this imbalanced myeloid activation remains a matter of debate. Mechanisms of immune evasion by SARS-CoV-2 have extensively been documented by Lei et al.⁵ The resulting delayed IFN signalling and sustained viral replication might promote cytolysis. This in turn might recruit immune cells and induce their cytokine production in a feedforward-loop manner⁶. A direct effect of SARS-CoV-2 proteins on epithelial and myeloid cells' cytokine production however was put forward in an elegant proteomic study, suggesting that SARS-CoV-2 nsp9 and nsp10 can interfere with NKRF (an NF-κB repressor) and stimulate IL-6 and IL-8 production^{4,7}. SARS-CoV and SARS-CoV-2, but not MERS-CoV, binding to and subsequent downregulation of ACE2 contributes to this pro-inflammatory signalling.⁸

However this remains a speculative debate for now and our study was not designed to address these questions, but rather aimed to characterize the immune dysregulation in critical COVID-19 with the clinical goal of rationalizing immunomodulatory treatments. We have added a paragraph on this topic to the “Discussion” (lines 348-369). We thank the reviewer for this suggestion.

6) Another major caveat, similar to a large number of recent literature is that this study relies heavily on the results obtained from PBMC. It is important to acknowledge this fact.

It is true that many of our analyses are PBMC-based, however we also present plasma cytokine, plasma NET formation as well as single cell RNA-seq data from bronchoalveolar lavage fluid from COVID-19 patients in this paper. Therefore, we added the following limitation to the discussion section (manuscript line 444-449): “Fourth, most of our analyses were performed on PBMCs, which not always reflect the ongoing disease processes at the site of infection. Nevertheless, these findings were complemented with plasma cytokine and NET biomarker measurements. Moreover, myeloid-driven inflammation was also evident from scRNA-seq data of COVID-19 BAL fluid, thereby expanding our observations to the local inflammatory compartment at the site of infection.”

REVIEWER II COMMENTS

Thank you for the opportunity to review Vanderbeke et. al. “Monocyte-driven atypical cytokine storm and aberrant neutrophil activation as key mediators of COVID-19 disease severity”. In this study, the authors employed a systems immunology approach to study differences in plasma inflammatory proteins and immune cell properties in patients with mild/moderate and critical COVID-19 disease, as well as patients with non-COVID19 Macrophage Activation Syndrome (MAS). The authors use a combination of analytical methods, including multiplex quantification of plasma protein concentration, analysis of immune cell distribution and function with mass cytometry and fluorescence flow cytometry, single-cell RNAseq of immune cells in PBMC and bronchoalveolar lavage, and analysis of NET formation. The authors also use an extensive array of bioinformatic methods to parse through the complex data thus generated, which yields interesting new findings regarding the role of monocyte and neutrophils in the pathobiology of COVID19. From a technical perspective, the proteomic, mass cytometry, and scRNA-seq data generated is impressive and appropriate measures are utilized to ensure data quality. My comments/concerns specific to the systems immunology approach are:

1) The authors generated a complex and multimodal dataset that provides a rich characterization of peripheral and local host immune response. However, it is unclear what the primary outcome of the study is, or what primary comparisons drive the study design. For example, in Fig. 1a the focus is on MAS vs COVID-19 patients, while Fig 1b is on COVID-19 patient severity. Similarly, Fig 2a shows a comparison of COVID-19 disease severity, while it is unclear what patient groups are being compared for Fig 2b. This is a major concern as additional group comparison increases the number of hypotheses tested significantly.

We reviewed the manuscript and figures thoroughly and agree that it was not always explained clearly which hypotheses and sample sets drive each analysis, nor which statistical tests have

been used to compare different cohorts. This sometimes clouds the narrative. We apologize for this and thank the reviewer for his comment. We have reanalyzed data and restructured results (in part based also on comments by Reviewers I and III), and have tried to highlight the research questions that formed the basis for specific experiments throughout the manuscript. The Methods section and figure legends have been updated (see also question 2). Below we provide a brief summary of our study design:

A first goal of our study was to identify similarities and differences between critical COVID-19 and MAS patients' chemokine and cytokine production, since multiple interventional trials are based on their (assumed) similarity (line 101-109). Because MAS patients are in critical clinical condition, we compared them to the subset of critically ill COVID-19 patients (in order to remove disease severity as a possible confounder). We also included a cohort of healthy controls, in order to grasp the magnitude of dysregulation. This analysis is shown in Fig. 1a and S1.

Secondly, we compared cytokine levels in mild and critically ill COVID-19 patients, again showing a healthy control cohort as a reference, as shown in Fig. 1b and S2. This analysis is relevant since immunomodulatory treatment is aimed at this critically ill subgroup of patients. As such, understanding what specifically drives critical disease, is of utmost importance (lines 134-135).

As a third step, we wanted to elucidate which immune cells orchestrate these cytokine changes. First, we performed immune prediction modelling (line 152-153) and then validated and characterized in-depth these immune cell populations using scRNA-seq, FACS and CyTOF (line 169-175). We identified classical monocytes as the main source of pro-inflammatory cytokines, and their relative increase (with a corresponding decrease of non-classical monocytes) to mark critical disease.

In a final step, we looked at downstream effects of these cytokine changes. First, based on the reduced levels of IFN- γ , we analyzed the quality of antigen presentation (lines 222-227). Next, we opted for an unbiased approach. Using correlation modelling, we identified a specific role for dysregulated neutrophils (line 244-247). This then urged us to perform plasma neutrophil activation and NETosis biomarker measurements as well as BAL fluid scRNA-seq analyses (lines 297-301) to validate the hypotheses generated by correlation modelling.

We have tried to better explain this flow throughout the manuscript, as indicated above, and believe this has improved the readability of the manuscript.

2) The statistical methods for evaluating patient group differences are not always obvious. While the authors describe in detail the computational approaches used for data processing and feature identification (for instance, multiple methods for immune cell identification and enumeration), insufficient information is provided for the reader to understand how patient cohorts are being compared to each other. For example, in Suppl. Fig 3, the statistical basis for the identification of differences in correlation networks between patient groups is unclear.

We apologize for this and have now carefully checked and adapted the Methods section (manuscript lines 512/517-521/592-608/683-685) as well as all figure and table legends accordingly (manuscript lines 975-976/983-984/991-992/996-997/1000-1001/1003/1027, suppl. information lines 20-21/48-49/90). We have also reanalyzed the data, by consistently applying multiple comparison corrections for the number of patient cohorts assessed.

Specifically, regarding the statistical methods utilized in Suppl. Fig. 3, we have used the following strategy. Firstly, a strict threshold of $\log_2(\text{fold-change})$ or $\log_2\text{FC}$ was applied for the ratio between healthy controls and the two specified subgroups of COVID-19 patients such that only those cytokines/chemokines were selected that had final $\log_2\text{FC} > 1$. A $\log_2\text{FC} > 1$ is considered to be a statistically sufficient threshold for defining differential changes in (expression) levels of proteins or genes⁹. This reference was added to the Methods section (line 512).

The statistical basis for the identification of differences in correlation networks between patient groups was based on iterative statistical thresholds tailored to each immune cell-type. These correlative networks were based on a background genetic profile generated from integration of already existing reference immune cell gene-expression profiles. More specifically, the (published) computational pipelines utilised for this analyses, assesses distribution of Pearson correlation coefficients in the true reference genetic-profiles of these immune cells and compares this with Pearson correlation coefficients obtained from a shuffled dataset (constructed with already known batch information available from the given datasets)¹⁰. The overall comparison of these true vs. shuffled Pearson correlation coefficients distributions served as a source of respective statistical thresholds based on primarily false discovery rates. These statistical thresholds per reference immune cell-type are now better indicated within the Methods section (lines 517-521).

3) Similarly, for the scRNA-seq data presented in Fig 2, it is not obvious whether groups of patients are being compared to one another, or whether the analysis focuses on differences between cell subsets. If it is the latter and the goal is to determine the cellular source of cytokines/chemokines increased in patients with COVID-19, it would be useful to know whether scRNA-seq data correlate with plasma cytokine concentration for the relevant cytokines.

The reviewer is correct to assume that this scRNA-seq analysis served the purpose of identifying the cellular source of cytokines. As such, gene expression of these cytokines between different cell subsets was compared, not between different groups of patients. This largely confirmed hypotheses generated by unbiased immune prediction modelling using experimental cytokine data. We have tried to state this more clearly in the manuscript (lines 169-175 and 183-188).

We fully agree with the relevance of a correlation analysis between immunophenotypic (scRNA-seq) data and plasma cytokine levels. Performing a correlation analysis between single-cell gene expression and bulk protein data however is not feasible, due to the data format incompatibility. Single cell RNA-seq is dense multi-variable data that cannot be simplified to a single value per patient, which is needed for such a correlation. Therefore, we used scRNA-seq to directly show expression of cytokine coding genes in the same group of patients in which cytokine levels were measured, but used our immunophenotypic mass cytometry data (which does provide a single value per immune cell subset per patient) to perform this type of correlation analysis on a single-cell level. We briefly added this to the manuscript (line 245-247).

4) The data provided in Fig. 4 (similarity matrix) provides only qualitative observations and a quantitative evaluation of between-group differences would be helpful.

Since the data source for the analyses presented in similarity matrices are extremely different in terms of data distribution scales and units, it is possible and advisable to integrate them on a qualitative level (since similarity matrices rely on correlations between entities and correlations by principle are independent of distribution scales and units as long as the data is sufficiently paired, as is the case herein). However, this is also a limiting factor for reliable quantitative analyses, since the patterns therein will be skewed simply due to technical rather than biological reasons.

As such, our research effort is structured so that the qualitative association studies serve as hypothesis-generating analyses, which are then followed by immunophenotyping experiments yielding quantitative data. We realize now this is not always clear from the manuscript, and have tried to point this out more explicitly in the revised manuscript (e.g. lines 153-154, 169-175, 245-247 and 297-301). We thank the reviewer for this remark.

5) With respect to the mass cytometry data, three approaches are utilized for cell cluster identification. While multiple approaches can be useful in evaluating the dependence of a finding on a particular cell-type identification method, the authors – to my knowledge – did not provide this information. A justification for the three approaches described in the methods and a comparative assessment would be useful.

The reviewer is correct in pointing out this issue. We now added the following clarification to the “Methods” section (lines 592-608): Different clustering methods can generate different results. Because of the lack of a gold standard, the unknown effects of COVID-19 on the immune system, and to avoid technical artefacts during clustering, we applied multiple clustering methods and followed a wisdom-of-the-crowds type of approach to identify the different immune cell populations. To perform a comparative assessment, we correlated the various approaches: FlowSOM vs Pathsetter, FlowSOM vs KMeans, KMeans vs Phenograph and FlowSOM vs Phenograph. Overall, all clustering algorithms concurred very well. To visualize this concurrence, we typically use a Jaccard similarity analysis where the various clustering methods are compared side-by-side. Below we show the plots of these comparisons based on the first tier of markers that were used to identify the overall immune cell populations. Using this approach, annotations were defined by a consensus-based approach where only those cells that agreed over at least 2 algorithms were pertained. As such, we observed that >92% of the cells showed agreement across all clustering methods, ~7% of the cells showed agreement in 2 or 3 clustering methods, and ~1% of the cells were discarded due to annotation disagreement.

Cluster Intersection: PhenoGraph vs FlowSom

Cluster Intersection: FlowSom vs KMeans

Figure R1. Heat map representing the Jaccard similarity index between the indicated clustering methods. This was optimised using a random subset of $1e^5$ cells following which the remainder of the cells were mapped on these clusters. The number of cells that were annotated to each cluster are indicated in the heatmap.

6) Some of the observed differences between patient cohorts could be driven by clinical or demographic variables other than the studied diseases. This is particularly important for the data presented in Fig. 1 where samples from COVID-19 patients are compared to historical samples. How did the author ensure that observed differences are not driven by confounding technical (sampling time), clinical or demographic variables?

The reviewer is correct in pointing out that potential sampling or demographic differences between study groups should be carefully assessed, especially when using historical controls.

For the prospectively recruited patient cohorts, we have summarized the demographic data in Table 1 and Supplementary Table 2 and now stress that the cohorts are well balanced. Methods are extensively described (lines 468-737): these procedures were identical for the “Healthy”, “Mild-moderate COVID-19” and “Critical COVID-19” cohorts as well as for “non-COVID-19 pneumonia” cases’.

For the comparison between the COVID-19 and Macrophage Activation Syndrome plasma cytokine profile, a historical cohort was indeed used. With respect to potential confounding technical variables, the MAS patients were recruited at the same University Hospital Leuven and their plasma was obtained by the same procedure as used in the present COVID-19 study. Namely, blood samples were gathered on admission and after separation, plasma was immediately placed in a -80°C freezer (as were the COVID-19 samples); ideal conditions for long time storage¹¹. However, we cannot completely rule out minor differences between MAS and COVID-19 cytokine levels due to a longer storage of MAS samples, and have acknowledged this in the “Limitations” section (line 440-443). Finally, cytokine analysis of plasma samples of all patient groups was simultaneously performed in the reaction plates.

Regarding clinical variables, MAS patients were in critical condition at the time of sampling. A comparison between the total COVID-19 cohort and MAS cohort, might indeed reveal differences based on different disease severity. Therefore, as also suggested by Reviewer I, we now performed a comparison between critically ill COVID-19 and MAS patients. Our most important findings, namely reduced type II interferon signaling, increased CXCL8 levels and increased VEGF comparing critical COVID-19 to MAS remain valid (manuscript lines 101-109).

Finally, with regards to demographic variables, the situation is more complex. MAS is primarily a pediatric condition. Expectedly, this study cohort has a lower median age and less (age-related) cardiovascular comorbidity, as well as fewer solid tumor malignancies. Since MAS is a primary immune dysfunction (as opposed to systemic immune dysregulation caused by a respiratory infection), the level of respiratory support in this cohort is also lower. The above stated demographic differences have now been included in the revised manuscript and are summarized in Table 1 and Supplementary Table 2.

Interestingly, in a recent study, Del Valle et al. performed a large-scale comparison of IL-1 β , IL-6, CXCL8 and TNF- α levels between healthy controls, (mild and critically ill) COVID-19 patients and multiple myeloma patients before and during Cytokine Release Syndrome (due to CAR T-cell Therapy). They conducted a robust multivariable regression analysis to correct for confounding demographic variables and found no influence of smoking status, COPD, HIV, sleep apnea, active malignancy, diabetes mellitus, arterial hypertension, congestive heart failure

or asthma on cytokine levels. In contrast, male gender (for IL-6), higher age (for IL-6, CXCL8, TNF- α), chronic kidney failure (for CXCL8, TNF- α) and atrial fibrillation (for IL-6, CXCL8) did augment cytokine levels¹².

We therefore also used linear regression analysis to correct for the influence of gender, age and chronic kidney failure on our key cytokine findings (as no patients in our study cohort had a diagnosis of atrial fibrillation). The significant differences for CXCL8, IFN- γ and VEGF all remained significant (see suppl. table 3).

However, as our study was underpowered to correct for these variables in a multivariable regression analysis, we cannot fully exclude an influence on our conclusions, which we have now acknowledged in the “Limitations” section of the manuscript (line 435-440). We thank the reviewer for his comment, yet we do want to additionally point out that our conclusions on cytokine shifts are strengthened by experimental evidence (i.e. scRNA-seq, CyTOF, FACS, NET biomarkers) of congruent changes in upstream and downstream immune cell subsets.

REVIEWER III COMMENTS

This manuscript presents a comprehensive analysis of immune correlates of moderate and severe SARS-CoV-2 infections. The authors use state-of-the-art immunology, RNA-seq and computational modelling to establish beyond doubt that COVID-19 patients show a “cytokine storm” that is phenotypically distinct from the canonical macrophage activation syndrome. Specifically, IFN- γ levels are much lower than expected and IFN- γ responses are much lower than in a “conventional” cytokine storm. This is driven in COVID-19 cases by monocytes and results in depressed antigen presentation and lowered Th1 and CD8+ cell levels. There is also evidence of a huge increase in neutrophil activation at sites of infection. Taken together these observations help explain the pathology of COVID-19 infection.

This is a very important manuscript that will help us understand COVID-19 pathology and may help with treatment regimes. It will also encourage the immunology community to think more broadly about using the dreadful phrase “cytokine storm” – it would appear that different pathogens can induce distinct types of response, and whilst perhaps not surprising, this should be more widely known. A clear demonstration using such a contemporary problem is to be welcomed.

I can't think of anything else they could have done to establish their case more clearly and this will undoubtedly be a work of reference in the field. It's not really a criticism but the manuscript is very descriptive; in trying to be as thorough (and non-committal) as possible it is a very “dry” read and will be impenetrable to anyone other than professional immunologists. For example, how many readers will know what “non-classical monocytes” are, and more importantly appreciate that these may play a significant role in antiviral defence and so the relative loss of these cells is probably very important in the disease? More narrative throughout the text would help non-experts to navigate. Similarly, although many will be happy with the concept that COVID-19 is predominantly a disease of immunopathology, as the authors go to great lengths to point out, this is an atypical (or at least uncommon) form of immunopathology for a respiratory infection; how does this square with the opening line of the summary – “host immune response to SARS-CoV-2, more so than viral factors, determines COVID-19 severity”? Surely because the immune response to

SARS-CoV-2 is distinct the virus must be in some way responsible for this, even if the pathological outcome is mediated by host immunity? The discussion would be improved by trying to contextualize the immunology in terms of the virus.

We thank the reviewer for these constructive though somewhat challenging comments, both regarding form and content of the manuscript. We have tried to improve the overall flow of the manuscript, by describing the reasoning behind the experiments and the logic of their sequence more clearly, as was also suggested by Reviewer II. We also integrated background immunological info throughout the manuscript for the non-professional immunologist reader.

Although our study was primarily designed to generate a multimodal immunological atlas of mild-moderate and critical COVID-19, the reviewer is correct to point out that findings should be contextualized within a virological framework. We have nuanced our opening statement in this way, now reading “Epidemiological and clinical reports have indicated that SARS-CoV-2 virulence hinges upon the triggering of an aberrant host immune response, more so than on direct virus-induced cellular damage.”

A balanced antiviral response can be broken down into two main parts, namely I) early IFN signalling followed by II) pro-inflammatory (IL-6, CXCL8, TNF- α) signalling. We have added a section on the physiology of this mechanism, based on work by Vabret et al. (lines 348-353)⁴. This “aberrant” host immune response we and others put forward consists of I) reduced/delayed IFN signalling with II) early and heightened pro-inflammatory signalling^{4,13}. A paragraph on mechanisms used by SARS-CoV-2 is now included in the “Discussion” (lines 353-367)⁵⁻⁷. Additionally, we refer to Galani et al. who have shown this immune dysregulation is not shared by influenza viruses¹³, which are major causes of severe viral pneumonia (lines 357-359). Other pathogenic coronaviruses SARS-CoV and MERS-CoV do have some of these IFN-response evasion and proinflammatory signalling mechanisms in common^{4,5,7} (lines 359-361). Finally, we briefly mention the differential effect of ACE2 on these mechanisms, as an entry molecule for SARS-CoV and SARS-CoV-2 but not MERS-CoV, as discussed by Ni et al. and Iwasaki et al.^{8,14} (lines 367-369).

References

1. Metzemaekers, M., Gouwy, M. & Proost, P. Neutrophil chemoattractant receptors in health and disease: double-edged swords. *Cell. Mol. Immunol.* **17**, 433–450 (2020).
2. Carvelli, J. *et al.* Association of COVID-19 inflammation with activation of the C5a–C5aRI axis. *Nature* **588**, (2020).
3. Vacchini, A. *et al.* Differential effects of posttranslational modifications of CXCL8/interleukin-8 on CXCR1 and CXCR2 internalization and signaling properties. *Int. J. Mol. Sci.* **19**, 1–18 (2018).
4. Vabret, N. *et al.* Review Immunology of COVID-19 : Current State of the Science. *Immunity*, (2020).
5. Lei, X. *et al.* Activation and evasion of type I interferon responses by SARS-CoV-2. *Nat. Commun.* **11**, 1–12 (2020).
6. Merad, M. & Martin, J. C. Pathological inflammation in patients with COVID-19: a key role for monocytes and macrophages. *Nat. Rev. Immunol.* **20**, 355–362 (2020).
7. Li, J. *et al.* Virus-Host Interactome and Proteomic Survey Reveal Potential Virulence Factors Influencing SARS-CoV-2 Pathogenesis. *Med* 1–15 (2020).
8. Iwasaki, M. *et al.* Inflammation Triggered by SARS-CoV-2 and ACE2 Augment Drives Multiple Organ Failure of Severe COVID-19: Molecular Mechanisms and Implications. *Inflammation* **44**, (2020).
9. Yip, S. H., Wang, P., Kocher, J. A., Sham, P. C. & Wang, J. Linnorm : improved statistical analysis for single cell RNA-seq expression data. **45**, 1–12 (2017).
10. Vandenbon, A. *et al.* Immuno-Navigator, a batch-corrected coexpression database, reveals cell type-specific gene networks in the immune system. *Proc. Natl. Acad. Sci. U. S. A.* **113**, E2393-2402 (2016).
11. De Jager, W., Bourcier, K., Rijkers, G. T., Prakken, B. J. & Seyfert-Margolis, V. Prerequisites for cytokine measurements in clinical trials with multiplex immunoassays. *BMC Immunol.* **10**, 52 (2009).
12. Del Valle, D. M. *et al.* An inflammatory cytokine signature predicts COVID-19 severity and survival. *Nat. Med.* **26**, 1636–1643 (2020).
13. Galani, I. E. *et al.* Untuned antiviral immunity in COVID-19 revealed by temporal type I/III interferon patterns and flu comparison. *Nat. Immunol.* **22**, 32–40 (2021).
14. Ni, W. *et al.* Role of angiotensin-converting enzyme 2 (ACE2) in COVID-19. *Crit. Care* **24**, 1–10 (2020).

REVIEWERS' COMMENTS

Reviewer #1 (Remarks to the Author):

The authors have carefully evaluated the comments, included re-analysed data, and satisfactorily addressed all the comments by this reviewer. As a results, the manuscript is significantly improved. This is a very good manuscript that will fill critical knowledge gap in our understanding of COVID19 pathogenesis. Great job by all authors involved!

Reviewer #2 (Remarks to the Author):

I have reviewed the revised manuscript “Monocyte-driven atypical cytokine storm and aberrant neutrophil activation as key mediators of COVID-19 disease severity”. The authors have addressed my comments by performing additional statistical analyses (e.g. comparison of clustering algorithms used in the CYTOF analysis, evaluation of potential confounding variables), clarifying the study design and statistical approaches (particularly specifying group comparisons), or referring to the limitations of the findings (e.g. technical limitations). The authors have performed an extensive and important work comparing the host response in the context of COVID-19 and MAS, and I have no further comment.

Reviewer #3 (Remarks to the Author):

The authors have addressed my concerns - thank you!